# T2L: Efficient Zero-Shot Action Recognition with Temporal Token Learning

**Shahzad Ahmad**                                                                      *shahzaa@hiof.no*
*Department of Computer Science and*
*Communication, Østfold University College, Norway*

**Sukalpa Chanda**                                                                     *sukalpa@ieee.org*
*Department of Computer Science and*
*Communication, Østfold University College, Norway*

**Yogesh S. Rawat**                                                                    *yogesh@ucf.edu*
*Center for Research in Computer Vision*
*University of Central Florida*

**Reviewed on OpenReview:** *https://openreview.net/forum?id=WvgoxpGpuU*

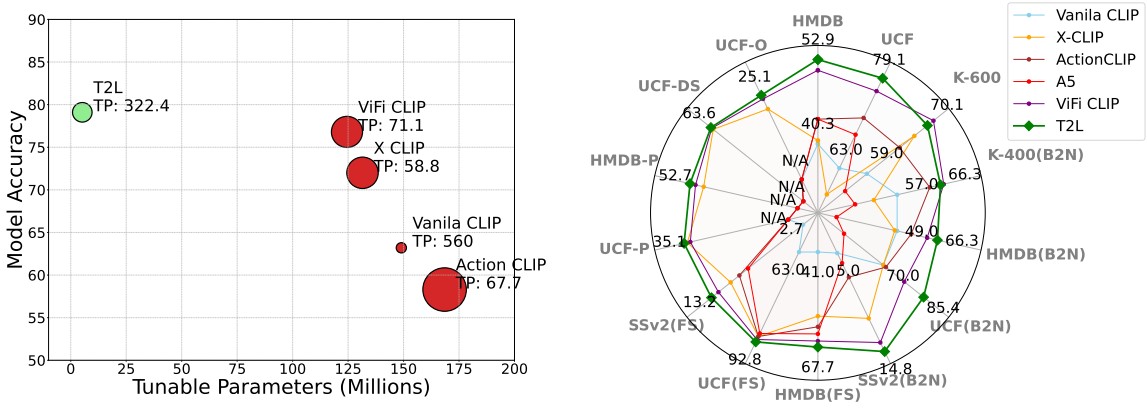

Figure 1: ***Effectiveness of T2L:*** (Left) T2L outperforms existing methods on the UCF-101 dataset for zero-shot evaluation with fewer tunable parameters and GFLOPS (bubble size), and higher throughput (TP). (Right) Evaluation across 14 different benchmarks demonstrating superior or competitive performance in comparison with existing approaches.

## Abstract

Recent advancements in large-scale pre-training of visual-language models on paired image-text data have demonstrated impressive generalization capabilities for zero-shot tasks. Building on this success, efforts have been made to adapt these image-based visual-language models, such as CLIP, for videos extending their zero-shot capabilities to the video domain. While these adaptations have shown promising results, they come at a significant *computational cost* and struggle with effectively *modeling the temporal aspects* inherent to the video domain. In this study, we present **Efficient Zero-Shot Action Recognition with Temporal Token Learning(T2L)**, a simple and ***efficient adaptation*** of CLIP that addresses these challenges. T2L leverages ***Temporal Token Learning (TTL)*** for seamless *temporal adaptation*, requiring no fundamental changes to the core CLIP architecture while preserving its remarkable *generalization abilities*. TTL relies on ***temporal feature diversity (TFD)***, a novel learning objective, which guides TTL to focus on *capturing motion*, thereby enhancing its learning capabilities from videos. We perform extensive experiments on ***nine*** different benchmark datasets, thoroughly evaluating T2L for zero-shot learning and base-to-novel video action

recognition, and also demonstrating its potential for few-shot generalization. Impressively, with merely 5.2 million learnable parameters, T2L can be efficiently trained on a single GPU *(with **25x less** learnable parameters, **3x reduction** in GFLOPs, and **4x improvement** in throughput when compared with prior best model)*, outperforming existing approaches in several evaluations. Code available here `https://github.com/Shahzadnit/T2L`.

# 1 Introduction

Large-scale pre-training of vision-language (VL) models on image-text pairs has shown exceptional effectiveness across diverse downstream tasks, particularly excelling in zero-shot scenarios Radford et al. (2021). Models like CLIP Radford et al. (2021) and ALIGN Jia et al. (2021), trained on vast internet-sourced data, provide strong, transferable representations with impressive generalization capabilities.

However, extending such a pre-training strategy to videos poses significant challenges. Unlike image-text pairs, aligned video-text data is scarce, and curating it is a challenging task Jia et al. (2021); Xu et al. (2021). Furthermore, videos inherently are more complex due to temporal dimension and entail substantial computational costs, while appearance cues can be captured efficiently through image-text pairs with a much lower compute budget. Therefore, adaptation of these image-language models for video-based tasks, while retaining their generic multimodal learned representations, is a promising research direction.

Motivated by this, recent works have adapted image-based VL models like CLIP Radford et al. (2021) for video representation learning by introducing additional learnable components for spatio-temporal modeling such as cross-frame self-attention layers Ju et al. (2022), textual or visual prompts Wang et al. (2021), or specialized video encoders Ni et al. (2022). More recently, Rasheed et al. (2023) proposed fine-tuning the entire CLIP model for video tasks. While these approaches achieve promising results, they are computationally intensive and often require fine-tuning large numbers of parameters, which can hinder generalization.

In light of these limitations, we propose T2L, an efficient adaptation of visual-language models for zero-shot action recognition. T2L introduces **Temporal Token Learning** coupled with a simple yet effective **motion constraint** to address the modeling of temporal aspects in videos. T2L does not update spatial features learned during pre-training, which preserves the generalization capabilities. The proposed approach is computationally efficient and can be trained on a single GPU with minimal learnable parameters.

We provide extensive evaluations across **nine** different action recognition benchmark datasets (Kinetics-400, Kinetics-600, UCF-101, HMDB-51, Something-something-v2, UCF-101-DS, UCF-101-P, HMDB51-P and UCF-101-O) for zero-shot, base-to-novel, and few-shot settings, showcasing our model's robust generalization capability, particularly in scenarios where motion plays a crucial role.

The main contributions of our work are as follows:

- We propose **efficient adaptation** of image-based visual-language models for zero-shot video action recognition.
- We introduce **Temporal Token Learning (TTL)** to model temporal aspect in videos. It effectively learns temporal dependencies across video frames with minimal learnable parameters.
- We propose a simple yet effective **temporal feature diversity (TFD)** loss which guides TTL in learning temporal behavior.
- We perform extensive evaluation across nine different action recognition datasets demonstrating its robust generalization capabilities. T2L outperforms previous best methods in several evaluations with merely 5.2 million learnable parameters.

# 2 Related Work

**Video Action Recognition** Effective video understanding requires modeling both spatial and motion cues. Vision Transformers have emerged as strong alternatives to 3D CNNs, capturing long-range spatio-temporal

dependencies and outperforming earlier models Feichtenhofer et al. (2019) Carreira & Zisserman (2017) Wang et al. (2017) Christoph & Pinz (2016). Moving beyond traditional uni-modal approaches, recent methods like ActionCLIP Wang et al. (2021), XCLIP Ni et al. (2022), and Ju et al. Ju et al. (2022) adopt a multi-modal framework, leveraging image-based visual-language models for zero-shot video understanding.

**Vision-Language Models** The effectiveness of learning multi-modal representations through large-scale image-text pre-training has been well established, showing strong performance across a wide range of uni-modal and multi-modal tasks Chen et al. (2020); Kamath et al. (2021); Li et al. (2019; 2020). Vision-Language (VL) models like CLIP Radford et al. (2021) and ALIGN Jia et al. (2021) have pioneered this area, using contrastive self-supervised objectives on large-scale image-caption pairs. These models exhibit impressive transfer capabilities in downstream vision tasks, including few-shot and zero-shot recognition, detection, and segmentation. However, extending these models to the video domain is challenging due to the absence of temporal cues in their image-level pre-training. To address this, recent efforts Ju et al. (2022); Ni et al. (2022); Wang et al. (2021) have adapted CLIP for video applications by integrating additional learnable modules—such as self-attention layers, visual or textual prompts, and dedicated visual decoders—showing performance improvements in video tasks. Specifically, Ju et al. Ju et al. (2022) adapt CLIP by incorporating text prompts and transformer layers for temporal modeling. While these temporal adaptations enhance performance in certain scenarios, they can reduce CLIP's generalization ability, particularly in zero-shot settings. To overcome these challenges, Rasheed et al. (2023) introduce ViFi-CLIP, a fully-supervised approach that improves both generalization and performance in zero-shot video understanding.

## 3 Methodology

We first discuss image-based visual language models (Sec. 3.1), then we introduce T2L (Sec. 3.2) and describe Temporal Token Learning (Sec. 3.2.1) followed by temporal feature diversity (Sec. 3.3).

### 3.1 Background

An image-based visual-language model comprises an image encoder $E_{\text{image}}$ and a text encoder $E_{\text{text}}$, jointly trained on large-scale image-text pairs by maximizing the similarity between image ($e_i$) and text ($y_i$) encodings from positive pairs. This is typically achieved using a contrastive objective Radford et al. (2021).

$$\mathcal{L}_c(e, y) = -\sum_{i=1} \log \frac{\exp(s(e_i, y_i)/\tau)}{\sum_{j=1}^{N} \exp(s(e_i, y_j)/\tau)} \tag{1}$$

where $s(e_i, y_i)$ is the cosine similarity between image encoding $e_i$ and its corresponding class encoding $y_i$, $N$ denotes the total number of classes, and $\tau$ is a temperature parameter applied to the similarity. This simple training strategy, when scaled to large datasets, has demonstrated impressive zero-shot capabilities. Once trained, these models can be applied to downstream tasks like image classification, where each target class is represented as a textual prompt using handcrafted templates such as '*this is a photo of <class name>*'. Classification is performed by matching the visual encoding with the textual prompts of candidate classes. This prompt-based representation enables the model's zero-shot capability.

### 3.2 T2L

Our goal is to efficiently adapt image-based visual-language models pre-trained on large-scale image-text pairs for video domain while preserving their zero-shot generalization capability. Since these encoders are pre-trained on image-text pair, they do not have any understanding for temporal aspects in videos. To overcome this challenge, existing methods have developed special temporal encoding blocks such as self-attention layers Ju et al. (2022) or dedicated video encoder modules Ni et al. (2022), which can learn the temporal relation between frames. However, such approaches are not very efficient in capturing the temporal cues since each video frame is encoded independently. Moreover, they require a large number of trainable parameters which is an overhead on training time and computing resources.

We want to adapt visual-language to capture temporal aspect from videos while preserving the spatial learning from images with minimal modifications (without changing the already learned weights). Towards this goal,

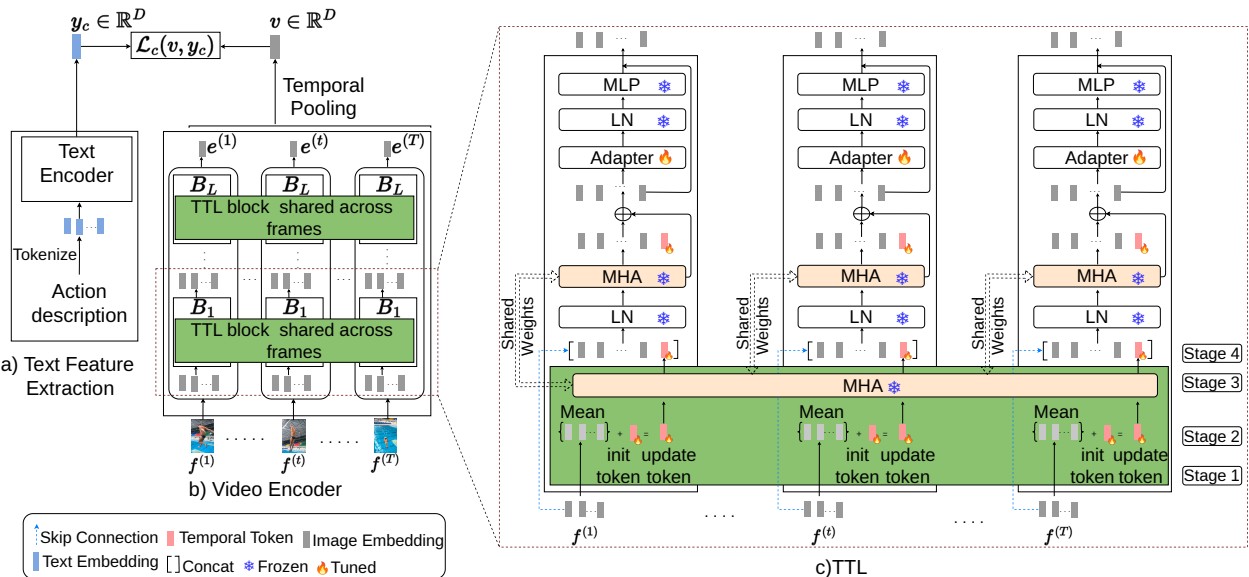

Figure 2: ***Overview of the proposed method:*** T2L utilizes Temporal Token Learning (TTL) to efficiently learn temporal aspects, eliminating the need for the frame integration module, a bottleneck in adapting image models for video understanding. Module ($a$) shows the text encoder(see detail in section 3.2.3). Module ($b$) depicts the architecture of a video encoder, where each frame is parallel encoded by a vision transformer. Temporal relations between frames are learned by module Temporal Token Learning(see detail in section 3.2.1)and within the vision transformer, each block is adapted using spatial adapters(see detail in section 3.2.2) shown in ($c$) .

we develop T2L, an Efficient Zero-shot video action recognition model based on CLIP Radford et al. (2021), which relies on Temporal Token Learning(TTL) and a novel temporal feature diversity (TFD) loss to capture motion cues from videos. We seamlessly integrate these elements into the CLIP architecture, leveraging its pre-trained capabilities. In this study, we experiment with CLIP but the proposed approach is general and should be applicable to other vision-language models.

**Problem formulation** Given a collection of video samples, we represent a video as $V = \{f^{(1)}, f^{(2)}, \ldots, f^{(T)}\} \in \mathbb{R}^{T \times H \times W \times C}$ consisting of $T$ frames represented as $f$, each with a resolution of $H \times W$ and $C = 3$ for RGB channels. Each video is associated with a text prompt $c$ which represents the target action class. We want a model $M_v$ which can leverage visual $E_i$ and text $E_t$ encoders from image-based pre-training and provide encoding for a video $V$ and its text-pair $c$ which are similar to each other and dissimilar from prompts of other action classes.

**Overview** Given a video with frame sequence, we rely on the image encoder $E_i$ for encoding each frame and the text encoder $E_t$ to encode the textual prompt for action class. The visual encoder handles each frame, resulting in the generation of frame-level embedding denoted as $\{e^{(1)}, e^{(2)}, \ldots, e^{(T)}\} \in \mathbb{R}^{T \times D}$ shown in Figure 2. Here, $e^{(T)}$ signifies the embedding of the $T^{th}$ frame within the video. The Temporal Token Learning help in capturing any relations between frames to model motion aspect in videos. The spatial features are also adapted to enable effective temporal learning. These frame embedding are then combined to create a holistic video-level representation $v \in \mathbb{R}^D$. The text encoder in image-based models also lack motion understanding therefore T2L also adapts text encoder along with visual encoder. During training, only temporal tokens, spatial adapters and text adapters are trained keeping all the weights from CLIP frozen.

### 3.2.1   Temporal Token Learning

Let's consider a video clip $V \in \mathbb{R}^{T \times H \times W \times C}$, composed of $T$ frames. Following a similar approach to the CLIP image encoder Radford et al. (2021), each $t$-th frame is divided into $N$ non-overlapping patches $\{f_i^{(t)}\}_{i=1}^N$ of size $\in \mathbb{R}^{P \times P \times C}$. Here, $t$ signifies the temporal index, and $N = \frac{H \times W}{P^2}$ where $P$ denotes the patch size. These patches, $\{f_i^{(t)}\}_{i=1}^N$, are then transformed into $D$-dimensional patch embedding $x_p^{(t)} \in \mathbb{R}^{N \times D}$. A class token

$x_{cls} \in \mathbb{R}^D$ is prepended to $x_p^{(t)}$ as $x_0^{(t)} = [x_{cls}; x_p^{(t)}] \in \mathbb{R}^{(N+1) \times D}$. To encode positional information, positional embedding $E_{pos} \in \mathbb{R}^{(N+1) \times D}$ are added to $x_0^{(t)}$ as $z_0^{(t)} = x_0^{(t)} + E_{pos}$, where $z_0^{(t)}$ is the final input being fed to a sequence of transformer blocks at the $t$-th frame.

To facilitate cross-frame temporal learning in a video, we introduce a novel concept called Temporal Token Learning.

**Stage 1: Initialization of Temporal Tokens**
These temporal tokens are strategically initialized at the input space of each Transformer layer, denoted as $P^{Temp} \in \mathbb{R}^{L \times T \times D}$, where $L$ is the number of layers in the CLIP Transformer. This collection of input learnable tokens is defined as $P^{Temp} = \{p_l \in \mathbb{R}^{T \times D} \mid l = 0, \dots, L-1\}$.

**Stage 2: Linking Temporal Tokens with Frame Embeddings**
Each $t$-th frame's embedding is linked to the respective temporal token as,

$$\tilde{p}_l^{(t)} = p_l^{(t)} + \frac{1}{N+1} \sum_{j=1}^{N+1} (z_{l-1}^{(t)})_j. \tag{2}$$

This is illustrated in Figure 2 module ($c$).

**Stage 3: Temporal Multi-Head Attention (MHA) Block**
To capture temporal dependencies effectively, we introduce an additional MHA operation within each transformer layer. This operation **shares the same parameters** as the existing MHA layer, ensuring consistency while maintaining computational efficiency. The shared MHA mechanism is dedicated to processing only the temporal tokens, enabling effective inter-frame modeling without increasing the number of trainable parameters.

**Stage 4: Processing Temporal Tokens**
Temporal tokens are obtained by aggregating the average pooling of each frame token and the learnable embeddings. Subsequently, MHA is performed on all tokens for each frame independently (including the temporal token). This operation at the $l$-th block is expressed as $\hat{p}_l = \text{MHA}(\text{LN}(\tilde{p}_l))$ where $p_l = [p_l^{(1)}, p_l^{(2)}, \dots, p_l^{(T)}]$, MHA represents a pre-trained and frozen attention block at the $l$-th layer derived from CLIP also shown in Figure 2 module ($c$), and LN denotes layer normalization. The final learned temporal token is concatenated with the frame embedding $[z_{l-1}^{(t)}, p_l^{(t)}]$ to facilitate further processing.

Our Temporal Token Learning approach is novel and unique as it enables efficient temporal learning without a separate architecture. By leveraging the existing CLIP image MHA with self-learnable tokens, we simplify the design while effectively capturing temporal dependencies, making it a highly efficient solution for temporal learning in video processing.

### 3.2.2 Spatial Adaptation

In Figure 2, module ($c$) depicts the transformer block structure $B_l$. After the Multi-Head Attention (MHA) module, the temporal token is discarded, and the frame patch token is forwarded for subsequent processing. The integration of the Adapter from Yang et al. (2023) after the self-attention layer within this transformer block modifies the CLIP architecture. For the image encoder, the computation within a transformer block is expressed as:

$$[\tilde{z}_l^{(t)}, \overline{p}_l^{(t)}] = [z_{l-1}^{(t)}, \hat{p}_l^{(t)}] + \text{MHA}(\text{LN}([z_{l-1}^{(t)}, \hat{p}_l^{(t)}])) \tag{3}$$

where $[\cdot, \cdot]$ concatenates the frame embeddings and temporal tokens. This output is then passed through an Adapter from Yang et al. (2023), followed by a residual block:

$$\hat{z}_l^{(t)} = \text{Adapter}(\tilde{z}_l^{(t)}) \tag{4}$$

$$z_l^{(t)} = \hat{z}_l^{(t)} + \text{MLP}(\text{LN}(\hat{z}_l^{(t)})), \quad l = 0, 1 \dots L-1 \tag{5}$$

**Gradient Flow for Temporal Token Learning** Gradient of the loss function $\mathcal{L}_{total}$ with respect to temporal tokens $P^{Temp} = \{p_l \in \mathbb{R}^{T \times D} \mid l = 0, \dots, L-1\}$ is shown in equation 6(See section A.1 in Appendix for detail derivation).

$$\frac{\partial \mathcal{L}_{total}}{\partial P^{Temp}} = \sum_{l=0}^{L-1} \sum_{t=1}^{T} \left( \frac{\partial \mathcal{L}_{total}}{\partial \overline{p}_l^{(t)}} \cdot \left(1 + \frac{\partial \text{MHA}(\text{LN}([z_{l-1}^{(t)}, \hat{p}_l^{(t)}]))}{\partial \hat{p}_l^{(t)}}\right) \cdot \frac{\partial \text{MHA}(\text{LN}(\tilde{p}_l))}{\partial \tilde{p}_l^{(t)}} \right) \tag{6}$$

### 3.2.3 Language Adaptation

To facilitate effective temporal learning, we adapt the text encoder to incorporate motion aspects into the textual prompts. Leveraging insights from efficient fine-tuning techniques in NLP Zhang et al. (2023); Zong et al. (2021); Zaken et al. (2021), recent advancements Zhang et al. (2023); Yang et al. (2023) have demonstrated strategies for fine-tuning without perturbing the original model weights. Specifically, we use the adapters from Zhang et al. (2023) to adapt the language model. We represent action class descriptions generated by a LLM (GPT-3.5) as textual prompts. We use a template *"describe [category] as an action performed by humans"* to generate descriptions. Throughout training, all transformer layers remain frozen except for the Adapters, which are updated. As shown in Figure 2, module (*b*), the text encoder adaptation is represented as $\tilde{y}_c = c_{des} + \text{MHA}(\text{LN}(c_{des}))$, $\hat{y}_c = \text{Adapter}(\tilde{y}_c)$, and $y_c = \hat{y}_c + \text{MLP}(\text{LN}(\hat{y}_c))$, where $c_{des}$ is the text embedding of the class description $c$.

### 3.3 Learning Objective

In the context of multimodal embedding models, our learning objective is to minimize the dissimilarity between video embeddings ($v$) and class embeddings ($y_c$) using a Contrastive loss Radford et al. (2021) (Eq. 1).

The contrastive loss function is vital for training, quantifying dissimilarity between predicted and ground-truth distributions to facilitate video and class embedding association. Yet, it may overlook video's intrinsic properties. For instance, if appearance suffices for classification, it might prioritize motionless video embeddings. To tackle this, we introduce the temporal feature diversity loss, emphasizing intrinsic motion properties, enhancing motion features in video embeddings.

**Temporal Feature Diversity** A video is composed of a sequence of $T$ frames picked at equal intervals. When there is motion in the video, generating embeddings $\{e^{(1)}, e^{(1)}, \ldots, e^{(T)}\}$ for each frame results in subtle differences among these embeddings. Our goal encompasses two facets: enhancing both the diversity (variance) and the distinctiveness (local diversity) among frame embeddings. This objective entails creating embeddings in a way that not only amplifies the differences between frames but also accentuates their central variations. To achieve this objective, we introduce the Temporal Feature Diversity (TFD) loss. Let $Var$ denote the degree of diversity among the frame embeddings, and let $C$ represent the measure of central difference. Then $Var \in \mathbb{R}^D$ and $C \in \mathbb{R}^D$ are defined as,

$$Var = \frac{1}{T} \sum_{i=1}^{T} (e^{(i)} - e_{\text{mean}})^2, \quad \text{where} \quad e_{\text{mean}} = \frac{1}{T} \sum_{i=1}^{T} e^{(i)} \tag{7}$$

where $(e^{(i)} - e_{\text{mean}})^2$ denotes element-wise squaring of the vector difference $(e^{(i)} - e_{\text{mean}}) \in \mathbb{R}^D$

$$C = \frac{1}{T} \sum_{i=1}^{T-1} \frac{\partial e^{(i)}}{\partial t}, \quad \text{where} \quad \frac{\partial e^{(i)}}{\partial t} = \frac{\|e^{(i+1)} - e^{(i-1)}\|}{2}. \tag{8}$$

$$\mathcal{L} = \frac{1}{dim(Var)} \sum_{i=1}^{dim(Var)} Var_i + \frac{1}{dim(C)} \sum_{i=1}^{dim(C)} C_i \tag{9}$$

Here $dim(Var)$ and $dim(C)$ is same as dimension of embeddings $e^i$. Our goal is to maximize the value of $\mathcal{L}$ during the training process, which amalgamates both the desired diversity and central distinctiveness. This leads us to the formulation of the Temporal Feature Diversity (TFD) loss:

$$\mathcal{L}_{\text{TFD}} = \frac{1}{\delta + \mathcal{L}} \quad \text{where} \quad \delta = 1. \tag{10}$$

Here, $\delta$ is a positive constant (specifically set to 1). The computed value of the loss is inversely proportional to the sum of $\delta$ and $\mathcal{L}$. This design choice emphasizes the importance of both high diversity and substantial central differences among frame embeddings. The overall learning objective is to minimize the final loss function termed as $\mathcal{L}_{total}$ that linearly combines the traditional contrastive loss $\mathcal{L}_c$(Eq. 1) along with $\mathcal{L}_{TFD}$ and is defined as,

$$\mathcal{L}_{total} = \lambda_1 \mathcal{L}_c(v, y_c) + \lambda_2 \mathcal{L}_{TFD} \tag{11}$$

where $\lambda_1$ and $\lambda_2$ are corresponding weights; we use equal weights for both in all our experiments.

### 3.4 Fine-Tuning Technique and Training Protocol

Our proposed method, Temporal Token Learning (T2L), efficiently models temporal dynamics in videos while maintaining computational efficiency. Unlike prior approaches that require extensive backbone fine-tuning (e.g., ActionCLIP Wang et al. (2021), X-CLIP Ni et al. (2022)), T2L introduces lightweight adaptations with a frozen base model.

**Frozen Backbone:** T2L leverages the pre-trained **CLIP ViT-B/16** (86.0M parameters) and the corresponding text encoder (63.0M parameters), keeping their 149.0M parameters frozen during training. This strategy preserves the generalization capabilities of the base model and significantly reduces computational overhead.

**Lightweight Adaptations:** T2L adds only 5.205M trainable parameters: **spatial adapters** (3.550M), **text adapters** (1.581M), and **temporal tokens** (0.074M). This constitutes just 3.5% of the total parameters, in contrast to prior methods that require extensive fine-tuning or architectural changes.

**Temporal Modeling:** T2L employs **TTL** for capturing temporal dependencies and **TFD loss** to enhance frame-level variability, achieving efficient temporal adaptation without additional temporal layers or self-attention mechanisms. Notably, TTL learns temporal consistency across all layers of the ViT-B/16 image encoder transformer, rather than relying solely on the output of the last layer.

**Efficiency:** By fine-tuning only adapters and temporal tokens, T2L minimizes FLOPs and memory usage while preserving high throughput efficiency. This lightweight approach achieves superior or competitive performance on nine benchmark datasets with minimal computational overhead compared to prior methods. This design underscores the efficiency and scalability of T2L, offering an effective solution for video understanding tasks.

## 4 Experiments and Results

**Datasets:** We evaluate our proposed method on nine different video action recognition benchmarks: Kinetics-400 Kay et al. (2017), Kinetics-600 Carreira et al. (2018), HMDB-51 Kuehne et al. (2011), UCF-101 Soomro et al. (2012), Something Something V2 (SSv2) Goyal et al. (2017), UCF-DS Schiappa et al. (2023), UCF-101-P, HMDB-51-PSchiappa et al. (2023), and UCF-101-O Modi et al. (2024). Kinetics-400, Kinetics-600, HMDB-51, and UCF-101 are known to have some appearance biases where background can also be important for recognizing actions Choi et al. (2019). On the other hand, Something-something-v2 is a more challenging dataset where temporal understanding is critical in recognizing the actions. UCF-DS, UCF-101-P, HMDB-51-P and UCF-101-O datasets have occlusion distribution shifts due to real-world perturbations instead of adversarial perturbations, making them useful for evaluating the robustness of models. Additional dataset details can be found in the Appendix A.5.

**Implementation Details:** We use ViT-B/16-based CLIP model as visual encoder in our experiments. In addition, we also assess our model's generalization using CLIP ViT-32 and CLIP ViT-14 backbones. The setup employs only 8 sparsely sampled frames per video, ensuring computational efficiency. We use AdamW optimizer with a base learning rate of $5 \times 10^{-5}$ to train out models for 50 epochs with and a weight decay of 0.2. The learning rate warms up for the initial 10% of epochs and then follows a cosine schedule. Training is performed on a single NVIDIA A100 80GB GPU, with a batch size of 70, and maintaining an input frame resolution of $224 \times 224$ pixels.

### 4.1 Evaluation and comparisons

Next, we show evaluation of T2L for zero-shot, base-to-novel, generalization capability for few-shot learning, and finally its robustness against distribution shift. Detail can be found in Appendix A.6.

**Zero-shot:** In this setting, the proposed model is trained on a source dataset, $D_S$ (Kinetics-400), and then directly tested on downstream cross-datasets, specifically HMDB-51, UCF-101, and Kinetics-600, for evaluation. The source dataset, $D_S$, contains samples from source classes, $Y_S = \{y_i\}_{i=0}^{k}$, and the evaluation is performed on the target dataset $D_T$, where $Y_S \cap Y_T = \emptyset$.

Table 1: **_Zero-shot comparison:_** We assess T2L against uni-modal approaches for zero-shot action recognition and image-based Vision-Language (VL) models adapted for video action recognition. Performance is measured using top-1 accuracy. Models are trained on Kinetics-400 and tested on HMDB-51, UCF-101, and Kinetics-600 (disjoint classes not shared with Kinetics-400).

| Method | Backbone | HMDB-51 | UCF-101 | K-600 | GFLOPS |
|---|---|---|---|---|---|
| **Uni-modal zero-shot action recognition models** | | | | | |
| ASR (Wang & Chen, 2017a) | BiDiLEL(Wang & Chen, 2017b) | 21.8 | 24.4 | – | – |
| ZSECOC (Qin et al., 2017) | Fisher Vectors(Perronnin et al., 2010) | 22.6 | 15.1 | – | – |
| UR (Zhu et al., 2018) | ResNet-200(Xie et al., 2017) | 24.4 | 17.5 | – | – |
| E2E (Brattoli et al., 2020) | R(2+1)D-18(Tran et al., 2018) | 32.7 | 48.0 | – | – |
| ER-ZSAR (Chen & Huang, 2021) | TSM(Lin et al., 2019) | 35.3 | 51.8 | – | – |
| **Adapting pre-trained image VL models** | | | | | |
| Vanilla CLIP (Radford et al., 2021) | ViT-16 | 40.8 | 63.2 | 59.8 | – |
| ActionCLIP (Wang et al., 2021) | ViT-16 | 40.8 | 58.3 | 67.7 | 563 |
| XCLIP (Ni et al., 2022) | ViT-16 | 44.6 | 72.0 | 65.2 | 287 |
| A5 (Ju et al., 2022) | ViT-16 | 44.3 | 69.3 | 55.8 | – |
| ViFi CLIP (Rasheed et al., 2023) | ViT-16 | 51.3 | 76.8 | **71.2** | 281 |
| T2L(ViT-16) | ViT-16 | **52.9** | **79.1** | 70.1 | 102 |
| T2L(ViT-32) | ViT-32 | 50.0 | 77.5 | 67.0 | 31.1 |
| T2L(ViT-14) | ViT-14 | 55.2 | 82.6 | 72.1 | 454.9 |

Table 2: **_Base to novel generalization:_** We perform comparison on four diverse datasets - Kinetics-400, HMDB-51, UCF-101, and SSv2. Models are evaluated using top-1 accuracy and HM is the harmonic mean of the base and novel classes.

| Method | Backbone | Kinetics-400 | | | HMDB-51 | | | UCF-101 | | | SSv2 | | |
|---|---|---|---|---|---|---|---|---|---|---|---|---|---|
| | | Base | Novel | HM | Base | Novel | HM | Base | Novel | HM | Base | Novel | HM |
| Vanilla CLIP (Radford et al., 2021) | ViT-16 | 62.3 | 53.4 | 57.5 | 53.3 | 46.8 | 49.8 | 78.5 | 63.6 | 70.3 | 4.9 | 5.3 | 5.1 |
| ActionCLIP (Wang et al., 2021) | ViT-16 | 61.0 | 46.2 | 52.6 | 69.1 | 37.3 | 48.5 | 90.1 | 58.1 | 70.7 | 13.3 | 10.1 | 11.5 |
| XCLIP (Ni et al., 2022) | ViT-16 | 74.1 | 56.4 | 64.0 | 69.4 | 45.5 | 55.0 | 89.9 | 58.9 | 71.2 | 8.5 | 6.6 | 7.4 |
| A5 (Ju et al., 2022) | ViT-16 | 69.7 | 37.6 | 48.8 | 46.2 | 16.0 | 23.8 | 90.5 | 40.4 | 55.8 | 8.3 | 5.3 | 6.4 |
| ViFi CLIP (Rasheed et al., 2023) | ViT-16 | **76.4** | **61.1** | **67.9** | 73.8 | 53.3 | 61.9 | 92.9 | 67.7 | 78.3 | 16.2 | 12.1 | 13.9 |
| T2L(ViT-16) | ViT-16 | 73.1 | 60.6 | 66.6 | **77.0** | **58.2** | **66.3** | **94.4** | **77.9** | **85.4** | **16.6** | **13.3** | **14.8** |
| T2L(ViT-32) | ViT-32 | 67.7 | 55.7 | 60.9 | 74.7 | 46.1 | 57.0 | 94.0 | 76.5 | 84.3 | 15.1 | 10.0 | 12.0 |
| T2L(ViT-14) | ViT-14 | 77.1 | 64.0 | 69.8 | 81.2 | 60.5 | 69.3 | 95.0 | 83.0 | 88.5 | 18.3 | 13.1 | 15.2 |

We compare T2L with both uni-modal models, and models adapting image-based multi-modal Vision-Language (VL) models (Table 1). T2L's distinguishing feature is its consistent performance, even with training on only 8 frames per video. This efficiency is due to its ability to learn appearance and motion harmoniously. This reduces computational demands and makes our model lightweight for streamlined training and deployment. As shown in Table 1, T2L outperforms existing models (with similar backbone ViT-16), achieving gains of +1.6% and +2.3% in HMDB-51 and UCF-101 respectively, whil showing competitive performance on Kinetics-600.

**Base-to-Novel:** To assess the generalization capabilities of T2L towards novel classes, we evaluate it in a base-to-novel setting Rasheed et al. (2023). We begin with a dataset $D_S$ with labels $Y_S = \{y_i\}_{i=0}^{k}$, which is partitioned into two categories: the base classes $Y_B$ and the novel classes $Y_N$. This partition ensures that $Y_B \cup Y_N = Y_S$ and $Y_B \cap Y_N = \emptyset$. The model is trained on the base classes using 16 samples per base class. The evaluation is shown in Table 2 for four diverse datasets: K-400, HMDB-51, UCF-101, and SSv2. We observe that T2L outperforms existing models on HMDB-51, UCF-101, and SSv2 under all evaluations with competitive performance on Kinetics-400.

**Generalization to few-Shot learning:** In the few-shot learning setting, we examine the model's capacity to learn under limited supervision. Given a dataset $D_S$ with labels $Y_S = \{y_i\}_{i=0}^{k}$, we create a general K-shot dataset where K samples are randomly selected from each category $y_i \in Y_S$ for training. We experiment with different values of K, specifically $K = 2, 4, 8$, and 16 and use the same samples as Rasheed et al. (2023). Evaluation is performed on the validation set of $D_S$. Table 3 shows the performance of T2L, alongside methods that adapt CLIP for video tasks. It's worth noting that while T2L doesn't exhibit significant performance improvements due to its relatively low number of tunable parameters, it consistently maintains good performance across different shot settings outperforming existing methods in most settings.

Table 3: ***Generalization to few-shot learning:*** Comparison of T2L with existing approaches on HMDB-51, UCF-101, and SSv2 Datasets across different few-shot scenarios (K = 2, 4, 6, 8 shots). Performance is evaluated using top-1 accuracy. Despite fewer tunable parameters, T2L exhibits robust generalization abilities, with improved performance across most evaluations.

| Method | Backbone | HMDB-51 | | | | UCF-101 | | | | SSv2 | | | |
|---|---|---|---|---|---|---|---|---|---|---|---|---|---|
| | | K=2 | K=4 | K=8 | K=16 | K=2 | K=4 | K=8 | K=16 | K=2 | K=4 | K=8 | K=16 |
| Vanilla CLIP (Radford et al., 2021) | ViT-16 | 41.9 | 41.9 | 41.9 | 41.9 | 63.6 | 63.6 | 63.6 | 63.6 | 2.7 | 2.7 | 2.7 | 2.7 |
| ActionCLIP(Wang et al., 2021) | ViT-16 | 47.5 | 57.9 | 57.3 | 59.1 | 70.6 | 71.5 | 73.0 | 91.4 | 4.1 | 5.8 | 8.4 | 11.1 |
| XCLIP (Ni et al., 2022) | ViT-16 | 53.0 | 57.3 | 62.8 | 62.4 | 71.4 | 79.9 | 83.7 | 91.4 | 3.9 | 4.5 | 6.8 | 10.0 |
| A5 (Ju et al., 2022) | ViT-16 | 39.7 | 50.7 | 57.0 | 62.4 | 71.4 | 79.9 | 85.7 | 89.9 | 4.4 | 5.1 | 6.1 | 9.7 |
| ViFi CLIP (Rasheed et al., 2023) | ViT-16 | 57.2 | **62.7** | 64.5 | 66.8 | 80.7 | 85.1 | 90.0 | 92.7 | 6.2 | 7.4 | 8.5 | 12.4 |
| T2L(ViT-16) | ViT-16 | **57.3** | 61.1 | **65.4** | **67.7** | **84.7** | **88.3** | **91.3** | **92.8** | **6.8** | **8.7** | **9.6** | **13.2** |
| T2L(ViT-32) | ViT-32 | 52.4 | 56.9 | 60.4 | 63.7 | 79.7 | 83.7 | 87.5 | 89.8 | 5.8 | 6.7 | 8.0 | 12.1 |
| T2L(ViT-14) | ViT-14 | 62.4 | 65.3 | 68.4 | 70.7 | 85.2 | 89.7 | 92.4 | 93.3 | 8.9 | 10.1 | 11.1 | 15.2 |

Table 4: ***Generalization to out-of-distribution datasets:*** Performance comparison of our model trained on the K-400 dataset and tested on extreme out-of-distribution datasets, including UCF-DS, UCF-101-P, HMDB-51-P, and UCF-101-O.

| Method | Backbone | HMDB-51-P | UCF-101-P | UCF-101-DS | UCF-101-O |
|---|---|---|---|---|---|
| ActionCLIP(Wang et al., 2021) | ViT-16 | 34.0 | 46.0 | 62.8 | 21.3 |
| ViFi CLIP (Rasheed et al., 2023) | ViT-16 | 33.1 | 50.6 | 63.1 | 24.8 |
| T2L(ViT-16) | ViT-16 | **35.1** | **52.7** | **63.7** | **25.1** |
| T2L(ViT-14) | ViT-14 | 36.6 | 55.7 | 71.9 | 27.7 |

**Robustness to distribution shift:** We evaluat the zero-shot performance of our model trained on the K-400 dataset, on extreme out-of-distribution datasets including UCF-DS Schiappa et al. (2023), UCF-101-P, HMDB-51-PSchiappa et al. (2023), and UCF-101-OModi et al. (2024). The results are summarized in Table 4, where we observed a comparatively better performance of the model, indicating its degree of robustness.

## 4.2 Ablations

In this section, we delve into the impact of Temporal Token Learning, temporal feature diversity loss, and their combined effect on our model's performance.

We use a base model with just spatial and text adapters as a baseline without Temporal Token Learning and temporal feature diversity loss. We perform this ablation on Kinetics-400, HMDB-51, UCF-101, and Something-something-v2 datasets.

**Impact of Temporal Token Learning:** We study the impact of Temporal Token Learning (TTL) by incorporating it into the baseline model, which is trained with spatial adapters at the image level. As shown in Tables 7, 8, and 10, TTL consistently improves accuracy across all datasets. The improvements are particularly notable on the SSv2 dataset compared to HMDB-51 and UCF-101, highlighting TTL's superior temporal modeling capability, especially for datasets with complex temporal dynamics. This demonstrates TTL's effectiveness in enhancing temporal representation and overall model performance.

**Impact of Temporal Feature Diversity Loss:** Adding the proposed TFD as a constraint during base model training consistently enhances performance, albeit to a lesser extent than observed with Temporal Token Learning in Tables 7 and 8. This improvement is attributed to the model's limited ability to learn temporal aspects, given the absence of temporal tokens.

**Impact of Temporal Feature Diversity Loss (TFD) on Video Frame Embeddings:** TFD plays a crucial role in enforcing diversity between frame embeddings within a video. Visualization of video frame embeddings using a Uniform Manifold Approximation and Projection (UMAP) McInnes et al. (2018) plot provides insights into the impact of TFD, as shown in Figure 3a. The UMAP plot, generated with 512-dimensional features reduced to 2 dimensions using parameters such as $n\_neighbors$=15, $min\_dist$=6, and $metric$='euclidean', highlights these effects. Without TFD, frame embeddings are densely clustered, suggesting the model predominantly focuses on appearance while neglecting motion. In contrast, with TFD, frame embeddings exhibit greater variance, reflecting the model's ability to discern motion by treating frames uniquely. These visualizations emphasize TFD's effectiveness in disentangling temporal features and improving motion understanding.

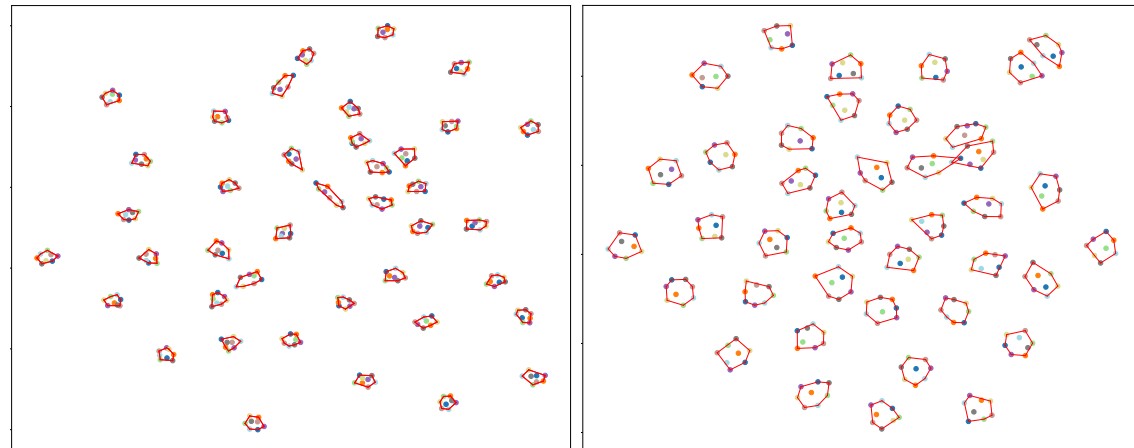

(a) UMAP plots for frame embeddings without(left) and with(right) TFD Loss

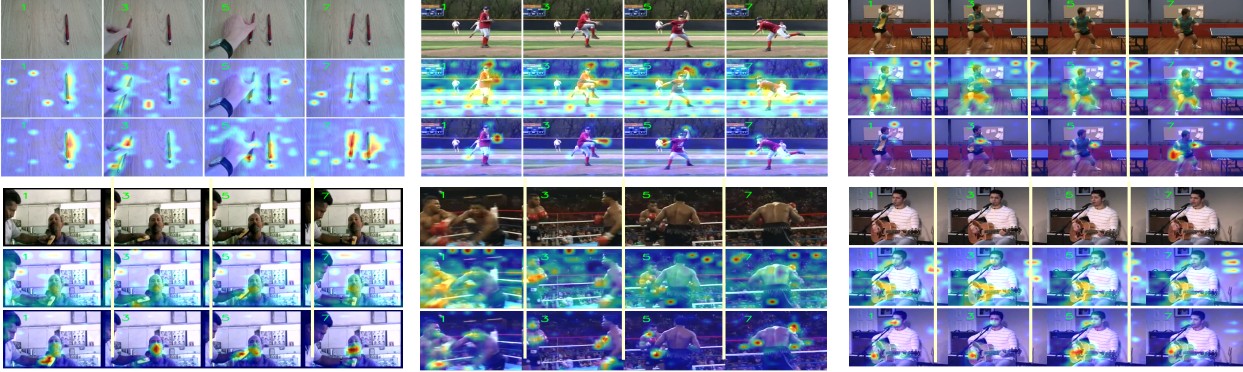

(b) Capturing motion dynamics within the video

Figure 3: **Subfigure** 3a: UMAP visualizations of frame embeddings, showcasing the effect of Temporal Feature Diversity (TFD). In the left plot, embeddings without TFD exhibit dense clustering, indicating limited feature separation. In contrast, the right plot demonstrates embeddings with TFD, revealing enhanced variance and better-defined clusters. Each cluster represents a single video, with individual dots inside the clusters corresponding to frames, highlighting how TFD disentangles temporal features effectively

**Subfigure** 3b Shows attention maps demonstrating the impact of Temporal Token Learning (T2L) and Temporal Feature Diversity (TFD) loss on SSv2 and UCF-101 validation sets. T2L effectively focuses on moving parts and objects, encoding video-specific information, as shown in the first example (*Putting something next to something*), where T2L highlights interactions and motion. The first row shows input frames, the second row shows base model ViT-16(with spatial adapters) results, and the third row shows TTL with TDF results. The attention heatmaps further capture motion dynamics by emphasizing subtle inter-frame changes, showcasing the model's ability to extract meaningful motion information.

**Additional Ablation Experiments:** We performed extensive ablation experiments to evaluate the effectiveness of various components of our approach. Specifically, we analyzed the performance of TTL and TFD in transitioning from base classes to novel classes in Table 8. For the few-shot setup, we examined the effectiveness of the LLM-generated descriptions on UCF-101, HMDB-51, and SSv2 in Table 9, and evaluated the effectiveness of TTL and TFD loss in Table 10.

**Fully-Supervised Comparison:** Table 5 provides a comparison of fully supervised results on the HMDB-51 and UCF-101 datasets. This comparison assesses the performance of T2L alongside uni-modal approaches for action recognition and image-based multi-modal Vision-Language (VL) models adapted for video action recognition. The evaluation metric used is top-1 accuracy. In this table, we observe the performance of T2L, a novel approach, compared to the baseline method, ActionCLIP Wang et al. (2021), on the HMDB-51 and UCF-101 datasets. Both methods utilize the same backbone architecture, ViT-16, and achieve competitive results.

Table 5: ***Fully-supervised comparison:*** Fully supervised results on HMDB-51 and UCF-101 datasets for T2L compared to uni-modal approaches for action recognition and image-based multi-modal Vision-Language (VL) models adapted for video action recognition. Performance is measured using top-1 accuracy. Red cells indicate fully supervised evaluation trained on respective datasets HMDB-51 and UCF-101, while blue cells indicate zero-shot evaluation. In the third row, zero-shot evaluation occurs when the model is trained on HMDB-51 for HMDB-P and on UCF-101 for zero-shot evaluation for UCF-101-P, UCF-101-O, and UCF-101-DS. †: Model trained on HMDB-51/UCF-101, ‡: zero-shot evaluation, model trained on Kinetics-400.

| Method | Backbone | HMDB-51 | UCF-101 | HMDB-51-P | UCF-101-P | UCF-101-O | UCF-101-DS |
|---|---|---|---|---|---|---|---|
| ActionCLIP (Wang et al., 2021) † | ViT-16 | 76.2 | 97.1 | - | - | - | - |
| ActionCLIP (Wang et al., 2021) ‡ | ViT-16 | 40.8 | 63.2 | 34.0 | 46.0 | 21.3 | 62.8 |
| T2L † | ViT-16 | 80.9 | 95.1 | 55.6 | 63.4 | 30.1 | 50.6 |
| T2L ‡ | ViT-16 | 52.9 | 79.1 | 35.1 | 52.7 | 25.1 | 63.7 |

Table 6: ***Zero-shot ablations on LLM description:*** Impact of LLM-generated action class descriptions as prompts on Zero-shot and Base-to-Novel settings.

| LLM-D | HMDB-51 | UCF-101 | K-600 | K-400 | | | HMDB-51 | | | UCF-101 | | |
|---|---|---|---|---|---|---|---|---|---|---|---|---|
| | Zero-shot | | | Base | Novel | HM | Base | Novel | HM | Base | Novel | HM |
| ✗ | 52.0 | 77.8 | 67.2 | 72.8 | 59.1 | 65.2 | 75.2 | 55.0 | 63.5 | 94.0 | 75.5 | 83.7 |
| ✓ | **52.9** | **79.1** | **70.1** | **73.1** | **60.6** | **66.3** | **77.0** | **58.2** | **66.3** | **94.4** | **77.9** | **85.4** |

Table 7: ***Zero-shot ablation on TTL and TFD:*** We evaluate the effectiveness of Temporal Token Learning (TTL) and temporal feature diversity loss(TFD) while using Zero shot setup train on Kinetics-400 tested on UCF-101, HMDB-51 and Kinetics-600.

| TTL | TFD | HMDB-51 | UCF-101 | K-600 |
|---|---|---|---|---|
| ✗ | ✗ | 49.2 | 75.5 | 63.1 |
| ✗ | ✓ | 50.0 | 75.9 | 65.5 |
| ✓ | ✗ | 51.2 | 78.1 | 67.5 |
| At first layer only | ✓ | 50.4 | 76.3 | 65.6 |
| ✓ | ✓ | **52.9** | **79.1** | **70.1** |

Table 8: ***Base to Novel ablation on TTL and TFD:*** We evaluate the effectiveness of Temporal Token Learning(TTL) and temporal feature diversity loss (TFD) using the base-to-novel setup on Kinetics-400, UCF-101, HMDB-51, and SSv2, using all samples from the base classes.

| Method | | Kinetics-400 | | | HMDB-51 | | | UCF-101 | | | SSv2 | | |
|---|---|---|---|---|---|---|---|---|---|---|---|---|---|
| TTL | TFD | Base | Novel | HM | Base | Novel | HM | Base | Novel | HM | Base | Novel | HM |
| ✗ | ✗ | 66.1 | 54.3 | 57.4 | 71.3 | 48.3 | 57.5 | 92.1 | 72.8 | 81.3 | 11.6 | 8.1 | 9.5 |
| ✗ | ✓ | 69.0 | 56.1 | 61.8 | 72.6 | 53.3 | 61.4 | 92.5 | 73.4 | 81.8 | 13.2 | 10.3 | 11.5 |
| ✓ | ✗ | 69.7 | 58.0 | 63.3 | 74.0 | 55.2 | 63.2 | 93.7 | 75.4 | 83.5 | 15.3 | 11.5 | 13.1 |
| ✓ | ✓ | **73.1** | **60.6** | **66.3** | **77.0** | **58.2** | **66.3** | **94.4** | **77.9** | **85.4** | **16.6** | **13.3** | **14.8** |

Table 9: ***Few shot Ablation LLM:*** Impact of LLM-generated action class descriptions as prompts on Few-shot setting.

| LLM-description | HMDB-51 | | | | UCF-101 | | | | SSv2 | | | |
|---|---|---|---|---|---|---|---|---|---|---|---|---|
| | K=2 | K=4 | K=8 | K=16 | K=2 | K=4 | K=8 | K=16 | K=2 | K=4 | K=8 | K=16 |
| ✗ | 57.0 | 59.1 | 65.1 | 65.2 | 81.8 | 86.6 | 88.8 | 90.2 | 6.1 | 8.0 | 9.3 | 11.9 |
| ✓ | **57.3** | **61.1** | **65.4** | **67.7** | **84.7** | **88.3** | **91.3** | **92.8** | **6.8** | **8.7** | **9.6** | **13.2** |

Specifically, T2L achieves a top-1 accuracy of 80.9% on HMDB-51 and 95.1% on UCF-101, outperforming ActionCLIP's accuracy of 76.2% on HMDB-51.

**Joint training:** Finally, when we train a model jointly with both Temporal Token Learning and temporal feature diversity loss, we observe significant improvement across all datasets; the improvement is significantly better in case of SSv2 dataset as compared with HMDB-51 and UCF-101. This further strengthens our claim that Temporal Token Learning and temporal feature diversity loss can help in better temporal modeling in videos.

## 4.3 Discussion and analysis

**UCF-101 and SSv2 as extremes**: UCF-101 and SSv2 represent the opposite ends of the spectrum in action recognition datasets. UCF-101 achieves high performance without explicit motion learning, leveraging

Table 10: ***Few shot ablation on TTL and TFD:*** We evaluate the effectiveness of Adaptive Temporal Token Tuning (TTL) and temporal feature diversity loss(TFD) while using Few-shot setup on UCF-101, HMDB-51 and SSv2.

| Method | | HMDB-51 | | | | UCF-101 | | | | SSv2 | | | |
|--------|-----|------|------|------|-------|------|------|------|-------|------|------|------|-------|
| TTL | TFD | K=2 | K=4 | K=8 | K=16 | K=2 | K=4 | K=8 | K=16 | K=2 | K=4 | K=8 | K=16 |
| ✗ | ✗ | 53.7 | 57.9 | 61.7 | 63.6 | 81.2 | 85.5 | 86.6 | 89.8 | 5.7 | 6.7 | 7.9 | 10.8 |
| ✗ | ✓ | 56.0 | 58.5 | 64.7 | 65.7 | 82.1 | 84.6 | 88.5 | 90.3 | 5.9 | 7.7 | 8.7 | 12.0 |
| ✓ | ✗ | 56.7 | 60.3 | 63.5 | 66.7 | 83.3 | 86.1 | 89.9 | 91.6 | 6.6 | 8.1 | 9.0 | 12.7 |
| ✓ | ✓ | **57.3** | **61.1** | **65.4** | **67.7** | **84.7** | **88.3** | **91.3** | **92.8** | **6.8** | **8.7** | **9.6** | **13.2** |

Figure 4: Comparison of T2L with ActionCLIP Wang et al. (2021), XCLIP Ni et al. (2022), and ViFi CLIP Rasheed et al. (2023). Throughput per view (TP) is measured on a single A100 GPU. T2L shows superior efficiency in GFLOPs, throughput, and parameter count, highlighting its computational advantages.

pre-trained CLIP weights that emphasize appearance. However, when Temporal Token Learning and temporal feature diversity loss are introduced, the attention shifts towards motion, as shown in the right part of Figure 3b. The baseline CLIP with adapters focuses on appearance and background, while T2L tracks the motion region. In contrast, SSv2, with its high reliance on motion cues, benefits significantly in our 'base to novel' experiments. Attention visualizations in Figure 3b highlight CLIP's emphasis on background, while T2L learns complex relative motion features.

**Effect of action class description:** Table 6 and Table 9 provides a detailed comparison of the model's performance with and without LLM-generated descriptions on two datasets: UCF-101 and HMDB-51. Our findings reveal that incorporating LLM-generated action class descriptions leads to improved text embedding generalization and enhanced model accuracy.

**Efficiency Analysis:** We analyze the computational complexity (see Figure 4) and observe that existing approaches show lower throughput due to added video-specific learnable components. In contrast, T2L effectively addresses this challenge by maintaining the base model parameters frozen, including 86.0M parameters for the CLIP ViT-B/16 Image Encoder and 63.0M parameters for the CLIP Text Encoder, resulting in a total of 149.0M frozen parameters. T2L adds only **5.2M** trainable parameters, consisting of **0.07M for Temporal Tokens**, **1.5M for Text Adapters**, and **3.5M for Spatial Visual Adapters**. This efficient design, with just 3.5% additional parameters over the base model, achieves higher throughput efficiency while maintaining comparable FLOPs to previous approaches, showcasing its ability to perform effective temporal modeling with minimal computational overhead.

## 5   Conclusion

In this work, we propose T2L, an efficient model for zero-shot video action recognition. T2L can effectively adapt image-based visual-language models to learn temporal aspect in videos while preserving its generalization capability. It is based on Temporal Token Learning which requires very few learnable parameters making the adaptation efficient. We also propose a novel temporal feature diversity loss which enhances the temporal modeling capability of T2L. With limited number of learnable parameters, T2L can be trained on a single GPU with significantly reduced computational resources, yet consistently achieving superior performance across different evaluations.

**Limitations:** In this work, we focused mainly on action recognition. However, learning the temporal aspects in videos is crucial for all video-based tasks. Exploring the benefits of the proposed approach for other video tasks, such as video summarizing, object tracking, and video segmentation, would be an interesting direction for future research.

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

# A Appendix

In this appendix section we present materials that enhances the understanding of our main paper through additional details and in-depth qualitative analysis. This appendix is structured as follows:

1. **Loss gradient flow for Temporal Token**

2. **Class Description Table**: A comprehensive table detailing the descriptions of various action classes, enhancing the clarity of our dataset.

3. **Dataset Details**: Detailed information about the datasets used in our study, shedding light on their characteristics and nuances.

4. **Evaluation Protocol**: A thorough explanation of our evaluation protocols, offering insights into our methodology and ensuring transparency in our assessments.

5. **Out-of-Distribution Results**: Further analysis of out-of-distribution results, demonstrating the robustness of our model in unconventional scenarios.

## A.1 Loss gradient flow for Temporal Token

To derive the gradient of the loss function $\mathcal{L}_{total}$ with respect to $P^{Temp} = \{p_l \in \mathbb{R}^{T \times D} \mid l = 0, \ldots, L-1\}$ at each layer of the transformer, we need to follow these steps:

Let's start with the key equations:

$$\tilde{p}_l^{(t)} = p_l^{(t)} + \frac{1}{N+1} \sum_{j=1}^{N+1} (z_{l-1}^{(t)})_j. \tag{12}$$

$$\hat{p}_l = \text{MHA}(\text{LN}(\tilde{p}_l)). \tag{13}$$

$$[\tilde{z}_l^{(t)}, \overline{p}_l^{(t)}] = [z_{l-1}^{(t)}, \hat{p}_l^{(t)}] + \text{MHA}(\text{LN}([z_{l-1}^{(t)}, \hat{p}_l^{(t)}])) \tag{14}$$

Gradient of the loss function $\mathcal{L}_{total}$ with respect to temporal tokens $P^{Temp} = \{p_l \in \mathbb{R}^{T \times D} \mid l = 0, \ldots, L-1\}$ is given

$$\frac{\partial \mathcal{L}_{total}}{\partial P^{Temp}} = \sum_{l=0}^{L-1} \sum_{t=1}^{T} \frac{\partial \mathcal{L}_{total}}{\partial p_l^{(t)}} \tag{15}$$

from Eq. 12, 13 and 14 we chain these gradients together using the chain rule to compute the gradient of the loss function $\mathcal{L}_{total}$ with respect to $p_l^{(t)}$:

$$\frac{\partial \mathcal{L}_{total}}{\partial p_l^{(t)}} = \frac{\partial \mathcal{L}_{total}}{\partial \overline{p}_l^{(t)}} \cdot \frac{\partial \overline{p}_l^{(t)}}{\partial \hat{p}_l^{(t)}} \cdot \frac{\partial \hat{p}_l^{(t)}}{\partial \tilde{p}_l^{(t)}} \cdot \frac{\partial \tilde{p}_l^{(t)}}{\partial p_l^{(t)}} \tag{16}$$

from Eq. 12 we can write

$$\frac{\partial \tilde{p}_l^{(t)}}{\partial p_l^{(t)}} = 1 \tag{17}$$

from Eq. 13 we can write

$$\frac{\partial \hat{p}_l^{(t)}}{\partial \tilde{p}_l^{(t)}} = \frac{\partial \text{MHA}(\text{LN}(\tilde{p}_l))}{\partial \tilde{p}_l^{(t)}} \tag{18}$$

from Eq. 14 we can write

$$\frac{\partial \overline{p}_l^{(t)}}{\partial \hat{p}_l^{(t)}} = 1 + \frac{\partial \text{MHA}(\text{LN}([z_{l-1}^{(t)}, \hat{p}_l^{(t)}]))}{\partial \hat{p}_l^{(t)}} \tag{19}$$

then,

$$\frac{\partial \mathcal{L}_{total}}{\partial p_l^{(t)}} = \frac{\partial \mathcal{L}_{total}}{\partial \overline{p}_l^{(t)}} \cdot \left(1 + \frac{\partial \text{MHA}(\text{LN}([z_{l-1}^{(t)}, \hat{p}_l^{(t)}]))}{\partial \hat{p}_l^{(t)}}\right) \cdot \frac{\partial \text{MHA}(\text{LN}(\tilde{p}_l))}{\partial \tilde{p}_l^{(t)}} \qquad (20)$$

Thus, the combined expression for the total gradient is:

$$\frac{\partial \mathcal{L}_{total}}{\partial P^{Temp}} = \sum_{l=0}^{L-1} \sum_{t=1}^{T} \left(\frac{\partial \mathcal{L}_{total}}{\partial \overline{p}_l^{(t)}} \cdot \left(1 + \frac{\partial \text{MHA}(\text{LN}([z_{l-1}^{(t)}, \hat{p}_l^{(t)}]))}{\partial \hat{p}_l^{(t)}}\right) \cdot \frac{\partial \text{MHA}(\text{LN}(\tilde{p}_l))}{\partial \tilde{p}_l^{(t)}}\right) \qquad (21)$$

## A.2 Class Description Table

In our pursuit of enhanced generalization, we utilized GPT-3.5 to generate action class names, aiming for a broader and more adaptable understanding of actions. To provide a comprehensive overview of our approach, we present a curated selection of UCF class names and their corresponding descriptions in Table 11. This table encapsulates the amalgamation of linguistic and visual comprehension, illustrating how our LLM-enhanced model interprets and defines various actions within the UCF-101 dataset.

Table 11: Description of selected UCF action classes generated by GPT-3.5.

| Action Class name | Example of GPT-3.5 Description |
|---|---|
| ApplyEyeMakeup | ApplyEyeMakeup is an action performed by humans to enhance their eyes and create a more dramatic look. It involves using a variety of makeup products such as eyeshadows, eyeliners, and mascaras, as well as blending and contouring techniques to create a desired effect. |
| Basketball | Basketball is a sport played by two teams of five players on a rectangular court. The objective is to shoot a ball through a hoop 18 inches in diameter and 10 feet high mounted to a backboard at each end. The game is played by bouncing the ball on a hard court surface. |
| PizzaTossing | Pizza tossing is an art form of spinning and stretching dough to create a thin and crispy pizza crust. It involves the use of hands, wrists, and arms to shape the dough into a round shape before it is placed in the oven. The goal is to achieve a consistent thickness and texture. |
| Typing | Typing is an action performed by humans using a keyboard to input text into a computer or other device. It is a skill that requires practice and repetition to become proficient. Typing can be used to create documents, send emails, type code, or enter data into databases. |

## A.3 Computational Efficiency and Performance Comparision

To provide a comprehensive evaluation of the efficiency and effectiveness of different design choices, we present a comparative analysis of FLOPs, throughput, trainable parameters, and accuracy in the tables below. This analysis compares three configurations: (1) Temporal Adapters, (2) Cross-Attention Fusion, and (3) Temporal Tokens (TTL).

The results in Tables 12 and 13 provide key insights into the trade-offs between computational efficiency and model performance for different temporal modeling approaches:

- **Computational Efficiency:** The T2L with Temporal Tokens (TTL) method achieves the lowest FLOPs (101.91 GFLOPs) while maintaining the highest throughput (322.0). Temporal Adapters introduce a higher computational cost (107.49 GFLOPs) with lower throughput (264.4).

- **Trainable Parameters:** TTL requires significantly fewer trainable parameters (0.074M) compared to Temporal Adapters (3.5M) and Cross-Attention Fusion (18.9M), demonstrating its efficiency.

Table 12: Comparison of FLOPs, throughput, and trainable parameters for different temporal modeling approaches. FLOPs are reported in GFLOPs.

| Method | Image Model | Text Model | Fusion Model | Total | Throughput | Trainable Params (M) |
|---|---|---|---|---|---|---|
| | FLOPs (GFLOPs) | | | | | |
| T2L with Temporal Adapters | 101.31 | 6.17 | N/A | 107.49 | 264.4 | 3.5 |
| T2L with Cross-Attention Fusion | 95.74 | 6.17 | 0.10 | 102.01 | 299.6 | 18.9 |
| T2L with Temporal Tokens (TTL) | 95.74 | 6.17 | N/A | 101.91 | 322.0 | 0.074 |

Table 13: Comparison of accuracy across different temporal modeling approaches.

| Method | Accuracy (%) | Novel Accuracy (%) | Harmonic Mean (%) |
|---|---|---|---|
| T2L with Temporal Adapters | 92.1 | 74.4 | 82.3 |
| T2L with Cross-Attention Fusion | 93.8 | 76.3 | 84.1 |
| **T2L with Temporal Tokens (TTL)** | **94.4** | **77.9** | **85.4** |

- **Performance Comparison:** While the primary focus of our paper is not a detailed base-to-novel evaluation here, this comparison is provided to illustrate the effectiveness of different temporal modeling choices. T2L with Temporal Tokens (TTL) consistently achieves the highest accuracy among the three approaches, with 94.4% overall accuracy.

- **Effectiveness of Temporal Learning:** Compared to Temporal Adapters and Cross-Attention Fusion, TTL provides the best balance of efficiency and accuracy. The significant reduction in trainable parameters and FLOPs, without sacrificing performance, demonstrates the effectiveness of the proposed method.

These findings validate that T2L with Temporal Tokens (TTL) offers the most efficient and effective approach among the compared temporal modeling methods.

## A.4 Impact of Discarding Temporal Tokens

To evaluate the necessity of retaining temporal tokens, we compare model performance when these tokens are kept vs. discarded.

**Motivation:** Temporal tokens encode global temporal relationships across frames using Multi-Head Attention (MHA) without adding significant parameter overhead. However, once they transfer temporal information to frame embeddings, retaining them may introduce redundancy.

**Efficiency Consideration:** Discarding temporal tokens optimizes computation while preserving accuracy, allowing the model to focus on refined frame-level embeddings.

Table 14: Comparison of retaining vs. discarding temporal tokens.

| Method | Accuracy (%) | Novel Accuracy (%) | Harmonic Mean (%) |
|---|---|---|---|
| T2L (Discard Temporal Tokens) | **94.4** | **77.9** | **85.4** |
| T2L (Retain Temporal Tokens) | 94.2 | 77.5 | 85.0 |

**Findings:** Table 14 shows that retaining temporal tokens provides **no significant benefit**. The harmonic mean remains nearly unchanged, validating our choice to discard them after their role in encoding temporal dependencies is fulfilled.

These results reinforce our design decision, demonstrating that temporal tokens efficiently propagate information before being removed. Future work could explore retaining CLS-style tokens for long-range temporal aggregation.

## A.5 Dataset Details

Our analysis is conducted on five well-established action recognition datasets: **Kinetics-400 and Kinetics-600 Carreira et al. (2018):** The Kinetics-400 dataset comprises 400 human action classes, represented by video clips sourced from various YouTube videos, each lasting around 10 seconds. It includes approximately 240,000 training videos and 20,000 validation videos. Kinetics-600 extends Kinetics-400, covering 600 action categories, with about 410,000 training video clips and 29,000 validation video clips.

**HMDB-51 (Kuehne et al., 2011):** The HMDB-51 dataset contains 71,000 realistic videos collected from various sources, spanning 51 action categories. The standard split includes 3,570 training samples and 1,530 validation samples. These sets are further divided into three splits, each containing 70 training clips and 30 validation clips for each action category.

**UCF-101 (Soomro et al., 2012):** UCF-101 comprises 13,000 realistic videos sourced from YouTube, encompassing 101 action categories, including five action types: human-object interaction, body-motion, human-human interaction, playing instrumental music, and sports. The standard split involves training on 9,537 videos and evaluation on 3,783 videos, distributed across three splits.

**Something Something V2 (SSv2) (Goyal et al., 2017):** The SSv2 dataset is an extensive collection of video clips depicting humans performing actions with everyday objects, spanning 174 action categories. This dataset focuses on recognizing fine-grained actions, such as covering something with something or uncovering something, making it more temporally biased compared to other datasets. The standard split consists of 168,913 training videos and 24,777 validation videos. Our reported performance metric is top-1 accuracy over the validation split.

**Out-of-Distribution Datasets:** Out-of-distribution datasets, such as UCF-DS Schiappa et al. (2023), UCF-101-P, HMDB-51-P Schiappa et al. (2023), and UCF-101-O Modi et al. (2024), play a crucial role in evaluating the robustness of action recognition models. These datasets focus on understanding how distribution shifts caused by real-world perturbations affect model performance, rather than solely considering adversarial manipulations. To achieve this goal, these datasets introduce four benchmark datasets, including HMDB51-P and UCF101-P, which encompass a diverse range of 90 perturbations. Additionally, UCF101-DS provides a platform for validating these findings by offering a dataset with realistic distribution shifts. Furthermore, UCF-101-O consists of benchmark datasets specifically designed to incorporate synthetically controlled static/dynamic occlusions. Collectively, these initiatives contribute to enhancing our understanding of model performance in challenging real-world scenarios.

## A.6 Evaluation Protocols

In our analysis, we explore four distinct experimental scenarios: zero-shot, base-to-novel generalization, few-shot, and fully-supervised settings. Across these scenarios, we employ a sparse sampling approach Wang et al. (2016), capturing 8 frames consistently. Each sampled frame is resized, with the shorter side set to 256 pixels, and centered to create a 224-pixel square crop.

### A.6.1 Zero-Shot Setting:

In the zero-shot setting, models trained on the Kinetics-400 dataset undergo evaluation on three different cross-datasets: HMDB-51, UCF-101, and Kinetics-600. For HMDB-51 and UCF-101, the methods are assessed across their respective three validation splits, and the top-1 average accuracy is reported. Regarding Kinetics-600, we assess the performance on 220 categories that do not overlap with Kinetics-400, reporting top-1 accuracy. In this setting, single-view inference using 8 frames is applied.

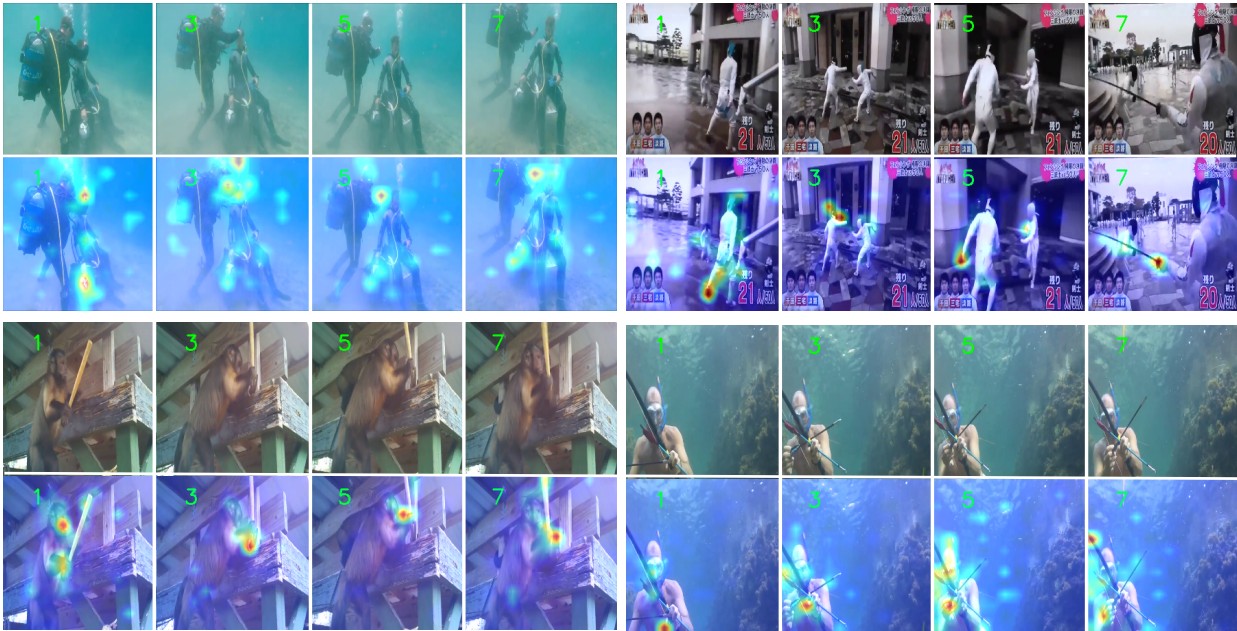

Figure 5: T2L's attention map visualizations on extreme out-of-distribution examples from the UCF-DS dataset Schiappa et al. (2023). The model demonstrates remarkable generalizability on unconventional actions like "Underwater Haircut" (top left), "Fencing in Crowd"(top right), "Hammering by Animal"(bottom left), and "Underwater Archery"(bottom right).

### A.6.2 Base-to-Novel Setting:

For a comprehensive assessment of the generalization capabilities of various approaches, we adopt the base-to-novel generalization setting Rasheed et al. (2023) for video action recognition tasks. Here, a model is initially trained on a set of base (seen) classes in a few-shot manner and subsequently evaluated on a set of novel (unseen) classes. We conduct a thorough generalization analysis across four datasets: Kinetics-400, HMDB-51, UCF-101, and SSv2. The dataset employs three training splits, classifying the total categories into two equal halves. The most frequently occurring classes constitute the base classes, while the rarely occurring categories are designated as the novel classes. This setup employs 8 frames and follows a single-view inference.

### A.6.3 Few-Shot Setting:

The few-shot setting involves creating a general K-shot split, with K samples used in accordance with splits from Rasheed et al. (2023). Specifically, we experiment with 2, 4, 8, and 16 shots on three datasets: HMDB-51, UCF-101, and SSv2. The models are assessed on the first validation split for HMDB-51 and UCF-101, and the full validation split, even on temporally-challenging datasets like SSv2.

### A.7 Out-of-Distribution Results

In our exploration of extreme out-of-distribution examples, we turned our attention to the UCF-DS dataset Schiappa et al. (2023), a collection of videos that push the boundaries of conventional contexts. These unconventional scenarios serve as litmus tests for a model's adaptability and robustness. In the figure:5, peculiar world of "Underwater Haircut", T2L's attention hones in on the person's head region, showcasing its ability to interpret unique actions. Venturing further, the model navigates the complex dynamics of "Fencing in a Crowd", deftly capturing the intricate hand motions amidst the chaos.Delving into more extraordinary territories, "Hammering by Animal" presents a scenario where an animal performs an unexpected action. T2L astutely focuses on the animal's hand motion, demonstrating its capacity to understand unconventional

movements . Finally, in the realm of "Underwater Archery",the model's attention locks onto the action area with precision, highlighting its exceptional adaptability. These insightful visualizations underscore T2L's extraordinary ability to navigate the unknown, positioning it not just as a tool for everyday scenarios but as a robust solution even in the face of the extraordinary .

