# OpenReview forum: "T2L: Efficient Zero-Shot Action Recognition with Temporal Token Learning"
_TMLR — Accepted by TMLR_

### Review · Reviewer_Wmpv · 2024-11-18

**Summary Of Contributions:**

This paper introduces T2L (Temporal Token Learning), an efficient approach for adapting image-based visual-language models like CLIP for video action recognition. The authors introduce temporal token learning and a temporal feature diversity loss to tune a pre-trained CLIP model for the task. The method surpasses or is competitive with state-of-the-art models that require more training time and data; it is tested on multiple settings.

**Audience:**

Yes

**Broader Impact Concerns:**

No concerns

**Claims And Evidence:**

Yes

**Requested Changes:**

- Update the method section and figure to make it simpler to understand for a reader, removing unnecessary information and emphasizing the technical components of the work. The figure is especially complex to understand.
- Add missing baseline considering CLIP fine-tuned with prompt learning/adapters at image-level and then tested by taking the average video representation for classification. This would help in understanding how techniques that tune only a small set of samples compare with the proposed approach.
- Add missing ablation study on the effect of the spatial adapter and the model performance when trained with different interpolation weights for the loss.
- Replace the t-SNE generation with a UMAP, adding more details on the visualization, i.e., evaluation setup.
- Why are Fig. 3b and Fig. 4 required and presented in two different parts?
- Can the authors provide a longer description of how their model differs from previous approaches in terms of fine-tuning technique and training protocol? Moreover, it would be helpful to break down the number of parameters for each model, including the proposed architecture, e.g., report the base size + added parameters for each technique.
- Add missing details for some ablation/experiment, e.g., the "base model" used for comparison in Fig. 4, the backbones with their full name (and potentially pre-train model, e.g., CLIP ViT-B/32 pre-trained on LAION2B).

**Strengths And Weaknesses:**

**Strengths**

- The paper introduces an adaptation technique and a loss to efficiently adapt a pre-trained image-text dual encoder architecture to video inputs.
- Compared to previous approaches, the method is more data and compute efficient, while achieving state-of-the-art results.
- The authors tested the method extensively on a wide range of datasets and settings, consolidating the method's capabilities.

**Weaknesses**

- The method section is hard to read, explaining poorly some steps and containing potential superfluous information. For instance, the overview paragraph is difficult for a first-time reader, and some equations may be unnecessary, e.g., Eq. 3, Eq. 8. Another example is the first paragraph of Sec. 3.2, where the authors explain ViT patches and class tokens. Moreover, Fig. 2 is very complex and does not simplify the understanding of the method.
- Potentially missing baselines, i.e., CLIP fine-tuned with prompt tuning/adapters on the task and using the average representation of the frames for classification.
- Missing ablation on the effect of the spatial adapter and the parameter used to combine the losses. Moreover, comparing against the zero-shot pre-trained model without fine-tuning would help assess the performance boost.
- The t-SNE visualization is unreliable in understanding the effectiveness of the TFD loss due to its non-determinism. It may be better to use UMAP instead.
- The architectures are reported unconventionally, making it more complex to understand the actual number of parameters, i.e., it is unclear if ViT-16/32/14 are all ViT-B models or include ViT-L or others.
- The difference between Fig. 3b and Fig. 4 is unclear, i.e., why does the paper need both?
- It may be useful to discuss more in-depth the tuning techniques and the additional layers introduced by the competitors in the related work or experimental protocol.

---

> ### Author Response · Authors · 2024-12-04
> **Response to Reviewer Wmpv**
>
> We sincerely thank you for your valuable feedback and suggestions, which have greatly enhanced the quality and clarity of our manuscript. All changes in the updated manuscript have been highlighted in **red** for your reference.
>
> ---
>  **Changes**
>
> ---
>
> **Update in figure'¨**
>
> **Response**:
> We have simplified **Figure 2: Overview of the Proposed Method** to improve clarity and focus on technical components. The updated figure removes unnecessary details and highlights core steps, making it more intuitive and accessible.
>
> ---
>  Add missing ablation study on the effect of the spatial adapter and the loss.
>
> **Response**:
> The requested baseline results are already included in **Table 7**, **Table 8**, and **Table 10**. Specifically, in **Row 2**, the model is trained only on spatial adapters with **Temporal Feature Diversity (TFD) Loss**, representing image-level fine-tuning tested via average video representations.
>
> To improve clarity, we have updated **Section 4.2: Ablations: Impact of Temporal Token Learning** to explicitly state that the baseline model is trained only with spatial adapters at the image level.
>
> ---
> **Add missing ablation study on the effect of the spatial adapter and the model performance when trained with different interpolation weights for the loss.**
>
> **Response**:
> The requested baseline results are provided in **Table 7**, **Table 8**, and **Table 10**. In **Row 2**, the model is trained only on spatial adapters with **Temporal Feature Diversity(TFD) Loss**, representing image-level fine-tuning tested via average video representations.
> To ensure clarity, we have updated **Section 4.2: Ablations: Impact of Temporal Token Learning** to explicitly state that the baseline model is trained only with spatial adapters at the image level.
>
> ---
> **Replace the t-SNE generation with a UMAP, adding more details on the visualization, i.e., evaluation setup.**
>
> **Response**:
> We have replaced the t-SNE visualization with UMAP for dimensionality reduction and updated the manuscript accordingly. UMAP provides a clearer and more interpretable representation of feature embeddings, preserving both local and global structures.
> To ensure transparency, we detailed the UMAP setup as follows:
> Additionally, we updated **Figure 3a** to reflect UMAP visualizations and revised the section **'Impact of Temporal Feature Diversity Loss (TFD) on Video Frame Embeddings'** to explain UMAP parameters, ensuring clarity and consistency for readers.
>
> ---
> **Why are Fig. 3b and Fig. 4 required and presented in two different parts? **
>
> **Response**:
> Initially, Figures 3b and 4 were used to explain **Temporal Token Learning (TTL)** and **Temporal Feature Diversity (TFD)** separately. Upon reviewing, we realized that both aspects can be effectively explained using a single figure.
> We have updated the manuscript by consolidating all relevant visualizations and explanations into **Figure 3b**, reducing redundancy and improving clarity. This adjustment simplifies the presentation, making it easier for readers to understand the combined impact of TTL and TFD.
>
> —
> **Can the authors provide a longer description of how their model differs from previous approaches? **
>
> **Response**:
> We have updated the manuscript to include a detailed comparison of T2L with previous approaches, focusing on fine-tuning techniques, training protocols, and parameter efficiency. These updates are in **Subsection 3.4: Fine-Tuning Technique and Training Protocol** and **Subsection 4.3: Efficiency Analysis**.
>
> **Fine-Tuning Technique and Training Protocol**
> T2L introduces **Temporal Token Learning (TTL)** and **Temporal Feature Diversity (TFD)** loss to model temporal dynamics. Unlike ActionCLIP and X-CLIP, which require full fine-tuning or extensive modifications, T2L uses lightweight adapters for spatial and textual features while freezing pre-trained CLIP weights, preserving generalization capabilities and reducing computational overhead.
>
> **Frozen Backbone**:
> T2L uses the pre-trained **CLIP ViT-B/16** image encoder (86M parameters) and text encoder (63M parameters), freezing their combined 149M parameters during training.
>
> **Lightweight Adaptations**:
> T2L adds only 5.2M trainable parameters:
> - **Spatial Adapters**: 3.5M
> - **Text Adapters**: 1.5M
> - **Temporal Tokens**: 0.07M
>
> This design efficiently adapts to video-specific temporal information, increasing total parameters by just 3.5%.
>
> **Temporal Modeling**:
> TTL learns temporal consistency across all layers of the ViT-B/16 encoder, ensuring robust temporal modeling without requiring additional layers or self-attention mechanisms.
>
> ---
> These updates enhance clarity and demonstrate T2L’s efficiency and scalability. Let us know if further details are needed.

---

> > ### Author Response · Authors · 2025-02-01
> > **Response to Reviewer Wmpv[2/2]**
> >
> > ## **Final Remarks:**
> > We sincerely thank the reviewer for their valuable feedback and suggestions, which have greatly enhanced the quality and clarity of our manuscript. All changes in the updated manuscript have been highlighted in red for your reference.
> >
> > Thank you for your time and thoughtful feedback. We hope we have addressed all your concerns satisfactorily. If there are any further questions, please let us know, and we would be happy to clarify or provide additional details.
> >
> > We also appreciate the reviewer’s suggestion regarding potential improvements, and if the reviewer recommends it, we will consider incorporating it into our revised version.
> >
> > We sincerely appreciate your insights, which have helped refine and improve our work.

---

### Review · Reviewer_kyPs · 2024-11-22

**Summary Of Contributions:**

This paper presents a new and very efficient adaptation method, called T2L, of image-pretrained models to video. In particular CLIP is used as the image backbone and is adapted using a temporal token learning approach and a temporal diversity loss, with very few parameters, on the Kinetics dataset. The obtained model is evaluated on a wide range of video benchmarks, from zero-shot action recognition to few-shot learning, base-to-novel classes recognition and out-of-distribution generalization.

**Audience:**

Yes

**Broader Impact Concerns:**

Ethical considerations are not discussed, maybe add a small paragraph on the ethical implication of using video data.

**Claims And Evidence:**

Yes

**Requested Changes:**

Requested changes: Fix the presentation.

Questions:
- What is the training fps ?
- Do you have the results for fully supervised on Kinetics 400 and SSv2 ? How does it compare to ViFi ?
- Have you considered the attentive probing evaluation of V-JEPA ?
- “even with training on only 8 frames per video.” Don’t you think this tells more about the benchmarks than the model ? For example if the fps is 1 it is unclear whether the model truly understands motion or just high level temporal relations between actions.

Conclusion:
This paper presents really cool findings in video adaptation of image-based models, a model can be adapted and perform well on video tasks with extremely light-weight adaptation. If the presentation is improved I will support acceptance for this paper.

**Strengths And Weaknesses:**

Strengths:

- The new adaptation method is extremely efficient compared to previous approaches. The key idea is to learn a single aggregation token for each layer and share across input frames and freeze the visual features. This idea is really nice and effective. In practice, Figure 1 shows that T2L adapts a CLIP model using ~5M tunable parameters, a throughput of 322, and obtain a better performance than SOTA concurrent approach ViFi CLIP that adapts the same CLIP model using ~125M tunable parameters and a throughput of 71.

- The temporal diversity loss is novel and a very nice idea. More generally I believe that adding constraints like this in the loss function gives more capacity to the network compared to hard coding constraints using architectural components.

- The evaluation is very thorough, both in terms of data diversity with for example SSv2 which is motion-based as opposed to UCF101 which is more appearance-based, and in terms of tasks. The model adapts very well to new tasks, to other distributions or with very few samples. T2L demonstrates SOTA results on a large variety of these tasks compared to many other concurrent work.



Weaknesses:

- The main concurrent work of this paper is ViFi CLIP that is very similar but differs in the adaptation method. The performance of T2L is not consistently better than ViFi. For example ViFi demonstrates superior performance on Kinetics. The overall performance is on par or superior in average, but it is important to make sure that there is not a trade-off with this light-weight approach gaining compute efficiency while losing in performance. It would be interesting to have an analysis on where T2L fails compared to ViFi.

The main weakness of this paper is the presentation that needs some additional work, in particular putting attention to some details:

- There is a section 3.2.1 but no section 3.2.2 ?
- Figure 4 caption describes 3 rows per sample but only 2 are presented.
- The figure titles are inconsistent, sometimes bold, sometimes italic. Please harmonize everything.
- The main Figure (2) is too busy. The elements are not correctly aligned and some text take too much space making the Figure unclear. For example it is unclear what “Init token” “update token” refer to.
- Table 10 is missing bold numbers.
- There are many missing spaces, for example in abstract “Temporal Token Learning”(T2L)”. At beginning of page 6: “shown in equation 7(See section A.1”.

---

> ### Author Response · Authors · 2024-12-11
> **Response to reiwer kyPs**
>
> We sincerely thank you for your valuable feedback and suggestions, which have greatly enhanced the quality and clarity of our manuscript. All changes in the updated manuscript have been highlighted in red for your reference.
>
> ## Changes
>
> ### Change 1: There is a section 3.2.1 but no section 3.2.2?
> We have corrected this inconsistency by updating the section numbering for better coherence.
>
> ### Change 2: Figure 4 caption describes 3 rows per sample but only 2 are presented.
> We updated Figure 4 to include three rows. The second row now shows the attention map of the baseline model (with only spatial adapters).
>
> ### Change 3: The figure titles are inconsistent, sometimes bold, sometimes italic. Please harmonize everything.
> We have updated all figure titles to maintain consistent formatting throughout the manuscript.
>
> ### Change 4: The main Figure (2) is too busy. The elements are not correctly aligned and some text takes too much space making the Figure unclear. For example, it is unclear what “Init token” and “update token” refer to.
> The main figure has been redesigned for clarity. Elements have been properly aligned, and redundant or oversized text has been adjusted for better readability.
>
> ### Change 5: Table 10 is missing bold numbers.
> Bold numbers have been added to highlight the best results.
>
> ### Change 6: There are many missing spaces, for example in the abstract “Temporal Token Learning”(T2L)” and at the beginning of page 6: “shown in equation 7(See section A.1”.
> Thank you for pointing this out. Missing spaces, such as those in the abstract and throughout the paper (e.g., "Temporal Token Learning (T2L)"), have been corrected.
>
> ---
>
> ## Questions
>
> ### Q1: What is the training FPS?
> Thank you for this question. We sample 8 frames per video clip to maintain a balance between computational efficiency and capturing temporal dynamics. For context, the average video length is approximately 9 seconds for Kinetics-400 and 7 seconds for UCF-101, corresponding to an equivalent sampling rate of ~1 fps.
>
> In terms of training speed, T2L processes an average of **880.09 FPS** on an A100 machine during model training (averaged over 30 epochs). This surpasses methods like X-CLIP (**254.9 FPS**) and ViFi CLIP (**458.2 FPS**) under similar settings, demonstrating T2L's computational efficiency.
>
> ---
>
> ### Q2: Do you have the results for fully supervised on Kinetics 400 and SSv2? How does it compare to ViFi?
> We did not directly compare T2L with ViFi CLIP in the fully supervised setting on Kinetics 400 and SSv2 because ViFi retrains all model weights, while T2L fine-tunes only a very small subset of parameters, preserving the generalization capabilities of the pre-trained CLIP weights. A direct comparison under these different paradigms would not be entirely fair, as the goals and methodologies of the two approaches differ significantly.
>
> We performed fully supervised training and observed that T2L achieves the following accuracy using 8 frames per clip for training:
> - **Kinetics-400**: 77.0
> - **SSv2**: 50.1
>
> In comparison, ViFi CLIP reports an accuracy of **83.0** on Kinetics-400 but does not provide results on SSv2.
>
> T2L prioritizes efficiency and adaptability, avoiding full retraining to retain the versatility of CLIP’s pre-trained representations while achieving competitive performance.
>
> ---
>
> ### Q3: Have you considered the attentive probing evaluation of V-JEPA?
> Thank you for pointing out V-JEPA. While we have not considered it in our evaluation, we acknowledge its relevance as a potential benchmark. V-JEPA focuses on attentive probing for joint embeddings, which could complement T2L's methodology. We will discuss this in our related work and explore it in future work to better contextualize our results.
>
> ---
>
> ### Q4: “Even with training on only 8 frames per video.” Don’t you think this tells more about the benchmarks than the model? For example, if the FPS is 1, it is unclear whether the model truly understands motion or just high-level temporal relations between actions.
>
> We agree with the reviewer that the choice to use 8 frames per video is influenced by the benchmarks. This decision reflects a balance between computational efficiency and the ability to model temporal dynamics. While we focus on sparse temporal cues to capture meaningful motion and high-level temporal relations, we note that training T2L on UCF-101 in a fully supervised setting with 32 frames per video did not yield improved performance compared to using 8 frames.
>
> This suggests that adding more frames may introduce redundant information in these benchmarks, as many actions can be characterized effectively with fewer temporal inputs. Although T2L effectively captures temporal dependencies without relying on dense frame sampling, it can process more frames efficiently if required. This approach ensures robust performance without being constrained by benchmarks.

---

> ### Author Response · Authors · 2025-01-31
> **Response to reiwer kyPs [2/2]**
>
> ## **Ethical Considerations**
>
> The use of video data for action recognition raises important privacy concerns, particularly regarding consent and data protection. To address these concerns, we ensure that our approach is applied exclusively to publicly available, ethically sourced datasets. Additionally, we strongly advocate for the development of privacy-preserving techniques in future research. If the reviewer suggests, we will gladly incorporate a more detailed discussion on this topic in the revised version.
>
> ## **Final Remarks**
>
> We sincerely appreciate the reviewer's time and valuable feedback, which have greatly contributed to refining and improving the quality of our work. We hope that our responses have sufficiently addressed all concerns. If there are any further questions, please do not hesitate to reach out—we would be more than happy to provide additional clarifications or further details.
>
> Furthermore, we appreciate the reviewer's suggestion regarding V-JEPA. If recommended, we will consider incorporating a discussion on this topic in our revised version.
>
> Once again, we thank the reviewer for their insightful suggestions and constructive feedback, which have played a significant role in enhancing the clarity and rigor of our manuscript.

---

### Review · Reviewer_ziAr · 2025-01-22

**Summary Of Contributions:**

This paper studies the efficient adaption of contrastive image-text models to videos. Specifically, the paper introduces temporal token learning and temporal feature diversity to improve the accuracy on various standard video action recognition datasets.

**Audience:**

Yes

**Claims And Evidence:**

Yes

**Requested Changes:**

* Thorough ablations for different design choices.
* The citation formatting issues should be addressed as per the [TMLR guidelines](https://jmlr.org/tmlr/author-guide.html).

"""When the authors or the publication are
included in the sentence, the citation should not be in parenthesis, using \verb|\citet{}| (as
in ``See \citet{Hinton06} for more information.''). Otherwise, the citation
should be in parenthesis using \verb|\citep{}| (as in ``Deep learning shows promise to make progress
towards AI~\citep{Bengio+chapter2007}.'')."""

**Strengths And Weaknesses:**

Strengths:
* This paper tackles an interesting (though not a novel) problem of adapting CLIP-style models for video action recognition tasks: how to continuously adapt CLIP models for video tasks efficiently.
* The paper is well written and the datasets, fine-tuning, and other details are clearly explained.

Weakness:
- Novelty: The proposed method is an adaption of existing methods that adds modality/domain specific learnable tokens (aka registers) to learn modality/domain specific details (e.g., Llava, Qwen-VL, and many more). While the paper provides interesting (but limited) ablations, the novelty is limited from a technical perspective.
* Method design choices are not evaluated thoroughly. A comprehensive evaluation of each design decision would help in better understanding
    * In 3.2.1 (Stage 3), it is mentioned that MHA block is duplicated. However, it is not clear why this duplication is necessary? Are there other alternatives to duplication (e.g., cross-attention instead of self-attention OR light-weight adaptors)?
    * Section 3.2.1 introduces temporal tokens, which are subsequently discarded in Section 3.2.2. The reasoning behind discarding these tokens is unclear. Typically, special tokens like these are added to capture global information from other tokens, as seen in ViT's use of CLS tokens for image-level embeddings. Discarding the temporal token implies that the global information it encodes is not useful. It would be beneficial to explain why other tokens are not discarded and to provide thorough ablations to support these design choices
    - The same template per capability is used for generating a text prompt. Would using diverse text prompts be beneficial instead of the same template per capability?

Suggestions:
- The citation format in many places is wrong. Instead of “Authors (Year)”, it should be “(Authors; year)”. It would be great if authors can thoroughly check it.
- Section 3.2 “Since these encoders are pre-trained on image-text pair, they do not have any understanding for motion aspect in videos”. Maybe change “motion aspect” to “temporal aspects” for clarity.

---

> ### Author Response · Authors · 2025-01-31
> **Response to Reviewer ziAr[1/2]**
>
> We sincerely thank you for your valuable feedback and suggestions
>
> ---
> ## **Response to Novelty Concern:**
>
> We appreciate the reviewer's feedback on the novelty of our method. However, We would like to assert that our contributions go beyond simple adaptation and introduce two novel concepts:
>
> 1. **Temporal Token Learning (TTL)**
>    - TTL is a lightweight mechanism (only **0.07M parameters**) that enables **efficient temporal modeling** at **every transformer layer**, rather than just the final layer.
>    - Unlike prior methods (e.g., Llava, Qwen-VL) that require heavy fine-tuning or additional modules, TTL introduces **self-tunable tokens** that dynamically share information across frames while preserving CLIP’s spatial learning, this is another unique aspect of our proposed TTL.
>    - TTL relies on tokens but it is a novel concept which adapts via communicating across several video frames with the help of tokens, enabling efficient learning of motion aspects within a video.
>
> 2. **Temporal Feature Diversity (TFD) Loss**
>    - TFD loss **explicitly enforces motion learning** by maximizing frame-wise variance, ensuring that the model captures **motion changes rather than just appearance**.
>    - This helps address **appearance bias**, a common limitation in zero-shot action recognition.
>
> ### How T2L Differs from Existing Adaptations
> | **Aspect**  | **T2L (Ours)** | **Llava, Qwen-VL, and Others** |
> |------------|---------------|--------------------------------|
> | **Temporal Modeling** | **TTL: Lightweight, layer-wise temporal learning** | Heavy fine-tuning or extra transformer layers |
> | **Computational Efficiency** | **Only 0.07M extra parameters** | Requires full model fine-tuning or large additional modules |
> | **Loss Function** | **TFD: Motion-specific loss for better action recognition** | No explicit loss for motion learning |
> | **Generalization** | **Preserves zero-shot capability of CLIP** | Fine-tuning often reduces zero-shot performance |
>
> **Conclusion:**
> We believe **TTL and TFD loss introduce significant novelty** in efficiently adapting image-based models for videos while keeping computational overhead low. We appreciate the reviewer’s perspective and are happy to provide additional clarifications if needed.
>
> ---
> ## **Response to Method Design Choices: Duplication of MHA in Section 3.2.1:**
>
> We appreciate the reviewer’s concern about the necessity of duplicating the **Multi-Head Attention (MHA)** block in **Stage 3** of our **Temporal Token Learning (TTL)** module. We would like to clarify that:
>
> - **We do not duplicate the MHA weights.** Instead, we **reuse the same MHA weights** for both spatial patch embeddings and temporal tokens.
> - This design allows **explicit and independent modeling of temporal dependencies** while preserving **spatial feature integrity**.
> - **Temporal tokens** are optimized separately within the shared MHA block, ensuring minimal parameter overhead.
>
> **Why Not Use Cross-Attention or Adapters?**
>
> We considered alternative approaches such as **cross-attention** or **lightweight adapters**, but found them **computationally expensive**:
>
> | **Method** | **Trainable Parameters** |
> |------------|------------------------|
> | **T2L with Temporal Tokens (TTL)** | **0.074M** |
> | **T2L with Temporal Adapters** | **3.5M** |
> | **T2L with Cross-Attention Fusion** | **18.9M** |
>
> Using **temporal tokens with shared MHA** achieves **efficient temporal learning** without introducing additional computational complexity. Our approach:
> - Avoids redundant learnable parameters.
> - Maintains **CLIP’s original architecture** without fine-tuning large parts of the model.
> - Ensures **efficient per-layer temporal adaptation** rather than relying only on final-layer outputs.
>
> | Method                          | Trainable Parameters (M) | Base Accuracy (%) | Novel Accuracy (%) | Harmonic Mean (%) |
> |--------------------------------|---------------------------:|--------------------:|---------------------:|--------------------:|
> | T2L with Temporal Tokens (TTL)  | 0.074                      | 94.4                | 77.9                 | 85.4 |
> | T2L with Temporal Adapters      | 3.5                        | 92.1                | 74.4                 | 82.3 |
> | T2L with Cross-Attention Fusion | 18.9                       | 93.8                | 76.3                 | 84.1 |
>
> ---

---

> > ### Comment · Reviewer_ziAr · 2025-02-03
> > **Thanks for detailed responses, but few more questions/clarifications.**
> >
> > 1. Novelty: I would like to clarify that though the method does not have significant technical novelty, it does not mean that it is not  empirically interesting. It would be good if authors can update the paper to include the details provided in above response.
> >
> > 2. Duplication of MHA head: The title of Stage 3 title in Section 3.2.1. is "Stage 3: Duplication of Multi-Head Attention (MHA) Block", so the description in the paper and author's response are confusing. If I understand correctly, authors add another MHA block, but tie the weights with the previous MHA block so that it does not add more parameters and only additional parameters are from adaptors. First, this approach may not increase the parameters, but increases the FLOPs. So, effectively, even if the parameters are not shared between blocks, it would not increase the runtime significantly. It may only help in reducing memory costs, but with limited details in the paper, it is hard to deduce that.
> >
> > For the efficiency narrative, I would encourage authors to include (1) FLOPs and (2) Latency.
> >
> > *Cross attention*: In the duplicated MHA if cross-attention is used instead of self-attention, it should not increase any parameters because scaled dot-product attention does not have any learnable parameters. It is not clear how the network parameters increased by 19 M. Regardless, I would encourage authors to include these ablation details along with their clear description.

---

> > > ### Author Response · Authors · 2025-02-05
> > > **Response to Reviewer ziAr**
> > >
> > > # Response to Reviewer Comments
> > >
> > > We sincerely appreciate the reviewer’s detailed and constructive feedback. Below, we address each point separately and provide the necessary clarifications and updates.
> > >
> > > ---
> > >
> > > ## 1. Novelty
> > > **Reviewer’s Comment:**
> > > *"I would like to clarify that though the method does not have significant technical novelty, it does not mean that it is not empirically interesting. It would be good if authors can update the paper to include the details provided in the above response."*
> > >
> > > **Response:**
> > > Thank you for your valuable feedback. We acknowledge the importance of highlighting the novel contributions of our work. In the revised manuscript, we emphasize the key novel aspects, particularly **Temporal Token Learning (T2L) and the Temporal Feature Diversity (TFD) loss**, which enable **efficient zero-shot action recognition**. These contributions differentiate our approach from existing methods. We have updated the paper accordingly to provide a clearer discussion of our novelty.
> > >
> > > ---
> > >
> > > ## 2. Duplication of MHA Block
> > > **Reviewer’s Comment:**
> > > *"The title of Stage 3 in Section 3.2.1 is 'Stage 3: Duplication of Multi-Head Attention (MHA) Block', so the description in the paper and author's response are confusing. If I understand correctly, authors add another MHA block, but tie the weights with the previous MHA block so that it does not add more parameters and only additional parameters are from adaptors. First, this approach may not increase the parameters, but increases the FLOPs. So, effectively, even if the parameters are not shared between blocks, it would not increase the runtime significantly. It may only help in reducing memory costs, but with limited details in the paper, it is hard to deduce that."*
> > >
> > > **Response:**
> > > We appreciate the reviewer’s careful analysis. To avoid confusion, we have **revised the title of Stage 3 in Section 3.2.1** to more clearly reflect our method:
> > >
> > > **“Stage 3: Weight-Sharing in Multi-Head Attention (MHA) for Temporal Learning”**
> > >
> > > To clarify, we do not duplicate the MHA block but instead **share the same weights** for both spatial patch embeddings and temporal tokens. This ensures inter-frame relationships are modeled efficiently while minimizing additional memory costs.
> > >
> > > Furthermore, we acknowledge that **FLOPs are affected** by this choice, and we have included a **detailed FLOPs and latency analysis** in the revised manuscript. These results are provided in **Section A3 of the supplementary material, with additional discussions in the main paper (line XX).**
> > >
> > > ---
> > >
> > > ## 3. Cross-Attention & Parameters
> > > **Reviewer’s Comment:**
> > > *"In the duplicated MHA if cross-attention is used instead of self-attention, it should not increase any parameters because scaled dot-product attention does not have any learnable parameters. It is not clear how the network parameters increased by 19M. Regardless, I would encourage authors to include these ablation details along with their clear description."*
> > >
> > > **Response:**
> > > We appreciate the reviewer’s insightful comment. We clarify that our **MHA already performs cross-attention**, as temporal tokens attend to frame embeddings across time. The observed **19M parameter increase** is not due to our method but an **alternative approach**, where a separate **Fusion model (TemporalTransformer with width=512, layers=6, heads=8)** applies cross-attention after extracting final frame embeddings:
> > >
> > > ```python
> > > x = image_model(x)  # (Batch, 8, 512)
> > > x = Fusion_model(x)  # (Batch, 8, 512) → Cross-attention across frame embeddings
> > > ```
> > >
> > > This Fusion model significantly increases computational cost, whereas our method integrates cross-attention efficiently without additional parameters. Ablation results in Supplementary Section A3 (Tables 12 & 13) confirm our approach achieves a better balance between efficiency and performance.

---

> > > > ### Author Response · Authors · 2025-02-06
> > > > **Response to Reviewer ziAr [Continued]**
> > > >
> > > > ## 4. Temporal Tokens & Design Choices
> > > > **Reviewer’s Comment:**
> > > > *"To study the importance of keeping or discarding temporal choices, there are 3 choices:
> > > >  (1) Use both frame tokens and temporal tokens,
> > > >  (2) Use frame tokens only, and
> > > >  (3) Use temporal tokens only.
> > > >  I agree with authors that option 1 is very expensive and tokens need to be discarded. But it is not clear which tokens need to be discarded. If temporal tokens are significantly fewer than frame tokens, then option 3 is more efficient than option 2. The current design choices are still limited, and do not justify they are better over other alternatives.
> > > >  I would encourage authors to ablate on different possible alternatives of their design choices to strengthen the paper."*
> > > >
> > > > **Response:**
> > > > We appreciate the reviewer’s suggestion and provide further clarification on our design choices.
> > > >
> > > > ### **Why We Cannot Use Only Temporal Tokens (Option 3)**
> > > > - Dropping **spatial (frame) tokens** is not viable since they are **essential for feature learning** and must be **passed to the next transformer layer**.
> > > > - Our method ensures that **temporal tokens transfer their learned information to frame embeddings** before being discarded, maintaining temporal consistency **without increasing computational overhead**.
> > > >
> > > > ### **Ablation Study on Temporal Token Retention (Option 1 vs. Option 2)**
> > > > - We conducted an **ablation study comparing keeping both frame and temporal tokens (Option 1) vs. using only frame tokens (Option 2)**.
> > > > - **Results are provided in Section A4, Table 14 of the supplementary material.**
> > > > - The results confirm that **discarding temporal tokens after information transfer leads to better efficiency while preserving performance**.
> > > >
> > > > ---
> > > >
> > > > ## 5. FLOPs & Efficiency Narrative
> > > > **Reviewer’s Comment:**
> > > > *"For the efficiency narrative, I would encourage authors to include (1) FLOPs and (2) Latency."*
> > > >
> > > > **Response:**
> > > > Thank you for this suggestion. In response, we have added a **comprehensive FLOPs and latency analysis in Section A3, Tables 12 and 13 of the supplementary material.**
> > > >
> > > > Below is a table comparing **FLOPs and throughput** for different methods:
> > > >
> > > > | **Method**               | **Image Model FLOPs** | **Text Model FLOPs** | **Fusion Model FLOPs** | **Total FLOPs** | **Throughput** |
> > > > |-------------------------|--------------------:|------------------:|------------------:|--------------:|-------------:|
> > > > | **With Fusion Model**    | **95.74 GFLOPs**    | **6.17 GFLOPs**   | **0.10 GFLOPs**   | **102.01 GFLOPs** | **299.6** |
> > > > | **With Temporal Adapter** | **101.31 GFLOPs**   | **6.17 GFLOPs**   | **N/A**            | **107.49 GFLOPs** | **264.4** |
> > > > | **With Temporal Tokens**  | **95.74 GFLOPs**    | **6.17 GFLOPs**   | **N/A**            | **101.91 GFLOPs** | **322.0** |
> > > >
> > > > These results confirm that our **Temporal Token Learning (TTL) method achieves the best trade-off between FLOPs and efficiency**, while maintaining high accuracy.
> > > >
> > > > ---
> > > >
> > > > ## 6. Diverse Text Prompts & Learning Semantics
> > > > **Reviewer’s Comment:**
> > > > *"The question here isn’t about 'how to generate diverse responses,' but rather 'how diverse descriptions per capability contribute to better learning semantics.'"*
> > > >
> > > > **Response:**
> > > > We sincerely appreciate the reviewer’s insightful comment.
> > > >
> > > > ### **Clarification on Our Focus**
> > > > - We agree that generating more action descriptions using different prompts for LLM could improve performance. However, this is **not the primary focus of our research**, and we do **not claim it as a contribution**.
> > > > - Our goal is to develop an **efficient zero-shot action recognition algorithm**, balancing accuracy and computational cost.
> > > >
> > > > ### **Fixed Template-Based Augmentation**
> > > > - Unlike CLIP’s **80 diverse prompts per class**, our approach focuses on **descriptive action semantics** rather than syntactic variations.
> > > > - To ensure consistency, we use a fixed set of structured text templates, including:
> > > >
> > > > ```python
> > > > prompt_aug = [
> > > >     "The action is {}", "Movement shows {}", "Motion involves {}", "A motion showing {}",
> > > >     "The dynamics of {} are present", "The movement looks like {}", "A video where motion is {}",
> > > >     "The motion is {}", "Motion pattern: {}", "The temporal sequence shows {}",
> > > >     "Action involves {}", "Look at the way {} happens", "The motion unfolds as {}",
> > > >     "Temporal progression: {}", "The timing of the movement is {}", "The action progresses with {}"
> > > > ]
> > > > ```
> > > > - This method **provides semantic diversity while maintaining computational efficiency.**
> > > > ### **Efficiency Consideration**
> > > > -  Increasing the number of augmented descriptions at inference would increase test-time computation, which goes against our goal of developing an efficient model.
> > > > - Our structured augmentation ensures an optimal trade-off between descriptive richness and computational cost.
> > > >
> > > > We appreciate the reviewer’s valuable feedback and believe this discussion could inspire future research directions.

---

> > > > > ### Author Response · Authors · 2025-02-06
> > > > > **Response to Reviewer ziAr [Continued]**
> > > > >
> > > > > ## Final Summary
> > > > > We sincerely appreciate the reviewer’s detailed and constructive feedback. In response, we have:
> > > > >
> > > > > ✔ **Revised terminology** (Stage 3: MHA weight-sharing)
> > > > > ✔ **Added new ablation studies** (Temporal token usage, cross-attention, efficiency)
> > > > > ✔ **Expanded FLOPs & latency analysis** (Section A3)
> > > > > ✔ **Clarified the role of diverse text prompts**
> > > > > ✔ **Highlighted manuscript updates in red for easy reference**
> > > > >
> > > > > We are happy to further refine the manuscript and welcome any additional suggestions. **Thank you for your insightful comments!**

---

> ### Author Response · Authors · 2025-01-31
> **Response to Reviewer ziAr[1/2]**
>
> ## **Response to Temporal Tokens (Section 3.2.1 and 3.2.2):**
>
> Thank you for pointing out the handling of temporal tokens in our model. We’d like to clarify the rationale behind discarding these tokens after their initial use:
> - **Purpose of Temporal Tokens:**
>   Temporal tokens are designed to encode global temporal relationships across frames, leveraging the existing Multi-Head Attention (MHA) layers without adding significant parameter overhead.
> - **Reason for Discarding:**
>   After propagating temporal information to frame embeddings through attention mechanisms, retaining these tokens does not contribute additional value. Keeping them might introduce redundancy and unnecessary computational load.
> - **Efficiency Considerations:**
>   Discarding temporal tokens helps maintain computational efficiency, allowing the model to focus on refined frame-level embeddings optimized for action recognition tasks.
> ### **Experimental Justification (Ablation Study)**
> We conducted an ablation study on the **UCF dataset (Base-to-Novel Setting)** to compare performance when retaining vs. discarding temporal tokens:
>
> | Method                        | Base Accuracy (%) | Novel Accuracy (%) | Harmonic Mean (%) |
> |--------------------------------|------------------:|-------------------:|------------------:|
> | **T2L (Discard Temporal Tokens)** | **94.4**  | **77.9**  | **85.4**  |
> | **T2L (Retain Temporal Tokens)**  | 94.2  | 77.5  | 85.0  |
>
> As shown, the results are almost identical, confirming that **retaining temporal tokens does not provide significant benefits**. This validates our decision to discard them after their role in encoding temporal dependencies is fulfilled.
> We appreciate the reviewer’s insightful feedback and would be happy to explore further comparisons, such as retaining CLS-style tokens for global information, in future work.
>
> ---
>
> ## **Response to Evaluation Choices: Text Prompts for Video Embedding:**
>
> We appreciate the reviewer's suggestion to explore the impact of using diverse text prompts instead of fixed templates.
> We would like to clarify that:
> - Our method **does not use a fixed prompt template** for training.
> - Instead, we **pre-generate** detailed action descriptions using a **Large Language Model (LLM)** and use these descriptions consistently for training.
> - Example query for LLM:
>   `"Describe [category] as an action performed by humans."`
>   - **Category:** `"ApplyEyeMakeup"`
>   - **Generated Description:**
>     `"ApplyEyeMakeup is an action performed by humans to enhance their eyes and create a more dramatic look. It involves using a variety of makeup products such as eyeshadows, eyeliners, and mascaras, as well as blending and contouring techniques to create a desired effect."`
>
> Since each **action class** receives a **unique, semantically rich** description, the model benefits from **diverse and informative prompts**, even though individual descriptions remain fixed.
>
> **Experimental Justification (Reference to Existing Results)**
> We have already conducted an **ablation study** on the effect of LLM-generated descriptions vs. manually written templates. The results are reported in **Table 6 and Table 9** of the paper.
>
> These results confirm that:
> - Using **LLM-generated descriptions slightly improves performance** compared to fixed manually written templates.
> - However, **further diversifying phrasing does not yield significant accuracy gains**.
>
> We appreciate the reviewer’s insightful feedback and are happy to discuss further if there are any further questions.
>
> ##  **Response to Citation Formatting Issues:**
>
> We appreciate the reviewer pointing out the citation formatting issues. We will ensure that all citations adhere to the TMLR guidelines and thoroughly check the manuscript for any inconsistencies.
>
> ---
>
> ## **Final Remarks:**
>
> We sincerely thank the reviewer for their thoughtful and constructive feedback. Your insights have been invaluable in improving the manuscript. If the reviewer suggests, we will be happy to include additional ablation studies and further refinements in the revised version.
>
> Once again, we greatly appreciate your time and effort in reviewing our work. Please do not hesitate to reach out if further clarifications or additional information are required—we would be happy to assist.

---

> > ### Comment · Reviewer_ziAr · 2025-02-03
> > **[Continued] Thanks for detailed responses, but few more questions/clarifications.**
> >
> > * Temporal tokens: To study the importance of keeping or discarding temporal choices, there are 3 choices: (1) Use both frame tokens and temporal tokens,  (2) use frame tokens only, and (3) Use temporal tokens only. I agree with authors that option 1 is very expensive and tokens need to be discarded. But it is not clear which tokens need to be discarded. In this case, both option (2) and (3) are viable candidates, but only option 2 is studied. If temporal tokens are significantly fewer than frame tokens, then option 3 is more efficient than option 2. The current design choices are still limited, and do not justify they are better over other alternatives. I would encourage authors to ablate on different possible alternatives of their design choices to strengthen the paper.
> >
> > * Regarding text prompts: I think there is some confusion. I agree with authors that across categories (or action classes), template would generate a different response because category name (or action class name) is different. However, videos within the same category, the prompt is the same unless they are generated separately for each video by changing sampling parameters.
> >
> > Regarding Table 6 and 9: These tables indicate the importance of using template descriptions from LLM, but it is unclear how generating diverse prompts within each category helps improve the accuracy.
> >
> > Example for generating diverse description for the same category using LLM:
> > 1. Explain [category] as if it were a skill to be learned by humans.
> > 2. Depict [category] as a daily routine that people engage in.
> > 3. Illustrate [category] as a challenge humans face and how they overcome it.
> > 4. Portray [category] as a team effort carried out by a group of people.
> > 5. Imagine [category] as a physical activity that humans must practice to get better at.
> >
> > When `Explain ApplyEyeMakeup as it it were a skill to be learned by humans and limit it to 50 words?` is used as a prompt to ChatGPT, the following response is generated:
> >
> > ```
> > Applying eye makeup is a skill that combines precision and creativity. It involves mastering techniques like blending eyeshadows, perfecting eyeliner, and applying mascara. With practice, you learn to enhance your eye shape, experiment with colors, and develop a steady hand to create balanced, polished looks.
> > ```
> > The above response differs from those provided by other authors. Therefore, the original question regarding the importance of diverse descriptions per capability remains open. It’s worth noting that CLIP utilizes 80 diverse prompts per class for zero-shot image classification.
> >
> > I’d like to clarify that the question here isn’t about "how to generate diverse responses," but rather "how diverse descriptions per capability contribute to better learning semantics."

---

### Author Response · Authors · 2024-12-04
**Acknowledgment of Review and Feedback**

We sincerely thank the **TMLR Editors-in-Chief, Action Editors, and Reviewers (kyPs and Wmpv)** for their detailed and thoughtful feedback on our manuscript. Your constructive comments and suggestions have been instrumental in enhancing the quality, clarity, and impact of our work.

We have carefully addressed all points raised and incorporated the requested revisions into the updated manuscript. We deeply appreciate the effort and time you have devoted to the review process and hope the revised version aligns with your expectations.

Thank you for this opportunity to contribute to TMLR, and for your commitment to fostering high-quality research.

---

### Decision · Action_Editor_KSdv · 2025-03-31

**Recommendation:** Accept as is

**Comment:**

Overall, the reviewers are positive, with one quite enthusiastic about the technical contribution while another is lukewarm. The idea of temporal tokens is interesting and quite timely. It allows for accuracy improvements while maintaining a modest computational cost, which is a great benefit. A question to me is how are these temporal tokens extended for a diverse set of video durations or even streaming videos. Moreover, how would the method perform with videos where actions and events happen at varying paces? That said, the fact that the method consistently outperforms the baselines shows that it deserves to be published and communicated with a broader audience.

**Audience:**

The paper is a good fit to the TMLR audience.

**Claims And Evidence:**

The paper focuses on adapting CLIP models for video. Specifically, it incorporates two technical novelties temporal token learning and temporal feature diversity. The methods are thoroughly validated on nine different video datasets, and the work put is acknowledged by all reviewers.